# TRIM28 regulates the nuclear accumulation and toxicity of both alpha-synuclein and tau

Maxime WC Rousseaux[1,2], Maria de Haro[1,2], Cristian A Lasagna-Reeves[1,2], Antonia De Maio[2,3], Jeehye Park[1,2†], Paymaan Jafar-Nejad[1,2‡], Ismael Al-Ramahi[1,2], Ajay Sharma[1,2], Lauren See[1,2], Nan Lu[1,2], Luis Vilanova-Velez[1,2], Tiemo J Klisch[1,2], Thomas F Westbrook[1,4], Juan C Troncoso[5], Juan Botas[1,2], Huda Y Zoghbi[1,2,3,6*]

[1]Department of Molecular and Human Genetics, Baylor College of Medicine, Houston, United States; [2]Jan and Dan Duncan Neurological Research Institute, Texas Children's Hospital, Houston, United States; [3]Program in Developmental Biology, Baylor College of Medicine, Houston, Canada; [4]The Verna and Marrs McLean Department of Biochemistry and Molecular Biology, Baylor College of Medicine, Houston, United States; [5]Division of Neuropathology, Department of Pathology, Johns Hopkins University School of Medicine, Baltimore, United States; [6]Howard Hughes Medical Institute, Baylor College of Medicine, Houston, United States

*For correspondence: hzoghbi@bcm.edu

Present address: †Program in Genetics and Genome Biology, The Hospital for Sick Children, The University of Toronto, Toronto, Canada; ‡Ionis Pharmaceuticals, Carlsbad, California, United States

**Abstract** Several neurodegenerative diseases are driven by the toxic gain-of-function of specific proteins within the brain. Elevated levels of alpha-synuclein (α-Syn) appear to drive neurotoxicity in Parkinson's disease (PD); neuronal accumulation of tau is a hallmark of Alzheimer's disease (AD); and their increased levels cause neurodegeneration in humans and model organisms. Despite the clinical differences between AD and PD, several lines of evidence suggest that α-Syn and tau overlap pathologically. The connections between α-Syn and tau led us to ask whether these proteins might be regulated through a shared pathway. We therefore screened for genes that affect post-translational levels of α-Syn and tau. We found that TRIM28 regulates α-Syn and tau levels and that its reduction rescues toxicity in animal models of tau- and α-Syn-mediated degeneration. TRIM28 stabilizes and promotes the nuclear accumulation and toxicity of both proteins. Intersecting screens across comorbid proteinopathies thus reveal shared mechanisms and therapeutic entry points.

## Introduction

A number of neurodegenerative diseases are caused by the gradual accumulation of specific proteins within neurons. As research on these 'proteinopathies' has progressed, certain unexpected commonalities have arisen. For example, alpha-synuclein (α-Syn, *SNCA*) is now thought to mediate neurotoxicity not only in Parkinson's disease (PD), but also in Parkinson's Disease Dementia (PDD) and Lewy Body Dementia (LBD) where it accumulates in Lewy bodies (*Spillantini et al., 1997*). Similarly, the microtubule associated protein tau (*MAPT*) accumulates not only in neurons affected by Alzheimer's disease (AD), but in PD and PDD, while mutations that increase the stability of tau result in frontotemporal dementia with parkinsonism (FTDP-17 (or FTD) [*Spillantini and Goedert, 2013*]). Coding region mutations in *SNCA* are by no means necessary for PD to develop given that duplication or triplication of the *SNCA* locus is sufficient to result in forms of PD whose onset and severity

**eLife digest** Behind many neurodegenerative diseases are specific proteins that abnormally accumulate inside neurons and damage the cells. In Parkinson's disease, the protein alpha-synuclein accumulates; in Alzheimer's disease, the protein tau is one of the toxic culprits; and in other neurodegenerative diseases, alpha-synuclein and tau both accumulate. Genetic studies suggest that accumulation of the two proteins may be linked, but little is known about the factors that regulate the levels of these proteins inside neurons.

Rousseaux et al. set out to identify how these proteins are regulated in the hope of finding new ways of targeting them and reducing their toxicity. Screening a subset of human genes led to one that encodes a protein called TRIM28, which regulates the levels of both alpha-synuclein and tau. When the TRIM28 protein was depleted in human and mouse cells, the levels of alpha-synuclein and tau also went down. This effect was specific because the levels of other proteins with the potential to cause neurodegeneration remained unaffected.

Models of neurodegenerative disease in fruit flies and mice were then used to explore how TRIM28 affects the levels of tau and alpha-synuclein in animals. In each case, the proteins' levels dropped when TRIM28 was suppressed and this in turn protected the neurons from damage. Rousseaux et al. went on to show that TRIM28 affected how alpha-synuclein and tau were cleared in cells. Overexpressing TRIM28 revealed that it could encourage both alpha-synuclein and tau to accumulate in the nucleus of cells over time.

Finally, Rousseaux et al. compared post-mortem brain tissue from people who had neurodegenerative conditions that are driven by or associated with tau and alpha-synuclein with tissue from those who did not. The cell nuclei in the diseased tissue had much more TRIM28 associated with alpha-synuclein and tau than those in the healthy tissues.

Overall, the findings show that TRIM28 promotes the accumulation and damaging effects of both alpha-synuclein and tau. The next steps will be to understand how TRIM28 does this. It will also be important to determine if this effect can be targeted, whilst leaving others roles of TRIM28 intact, in order to explore it as a potential target to treat or prevent neurodegenerative diseases.

correlate with gene dosage (*Chartier-Harlin et al., 2004*; *Ibanez et al., 2004*; *Singleton, 2003*). Supporting these clinical findings are studies that show that overexpression of wild-type forms of either *SNCA* or *MAPT* elicit neurodegeneration in model organisms, whereas suppressing their levels appears to be neuroprotective (*Dauer et al., 2002*; *Ishihara et al., 1999*; *Jackson et al., 2002*; *Rapoport et al., 2002*; *Rockenstein et al., 2002*; *Wittmann, 2001*). Given the brain's sensitivity to the dosage of either of these proteins, one would expect elevated levels of multiple proteins to be even more problematic, and this proves to be the case (*Moussaud et al., 2014*).

Pathogenic proteins involved in different proteionopathies often interact with each other and cause cellular toxicity, either due to their additive effects on downstream activities or further compromise of protein homeostasis (*Clinton et al., 2010*). Abnormally aggregated α-Syn and tau are often found together in postmortem cases of PD and LBD (*Arima et al., 1999*; *Colom-Cadena et al., 2013*; *Iseki et al., 2002*; *Ishizawa et al., 2003*), and genetic interaction studies in *Drosophila* demonstrate that α-Syn and tau synergize in promoting toxicity (*Roy and Jackson, 2014*). Biochemical evidence even suggests that α-Syn may act as an amyloidogenic 'seed' for the accumulation of tau, and vice versa (*Guo et al., 2013*; *Lasagna-Reeves et al., 2010*; *Sengupta et al., 2015*). Even more, several genome-wide association studies have reported genetic interaction between tau and alpha-synuclein in PD pathogenesis (*Simón-Sánchez et al., 2009*). Thus, the interaction between tau and α-synuclein is gaining increased attention for its possible pathogenic role in synucleinopathies and tauopathies. The mechanisms governing the dual accumulation of α-Syn and tau remain elusive, but it seems plausible that decreasing levels of either or both of these proteins could prove an effective therapeutic strategy for this family of diseases.

Inspired by the extensive overlap between α-Syn and tau pathology and their corresponding clinical phenotypes (*Galpern and Lang, 2006*), we reasoned that they may be regulated through shared pathways and that dysfunction of these regulatory pathways may lead to their pathogenic

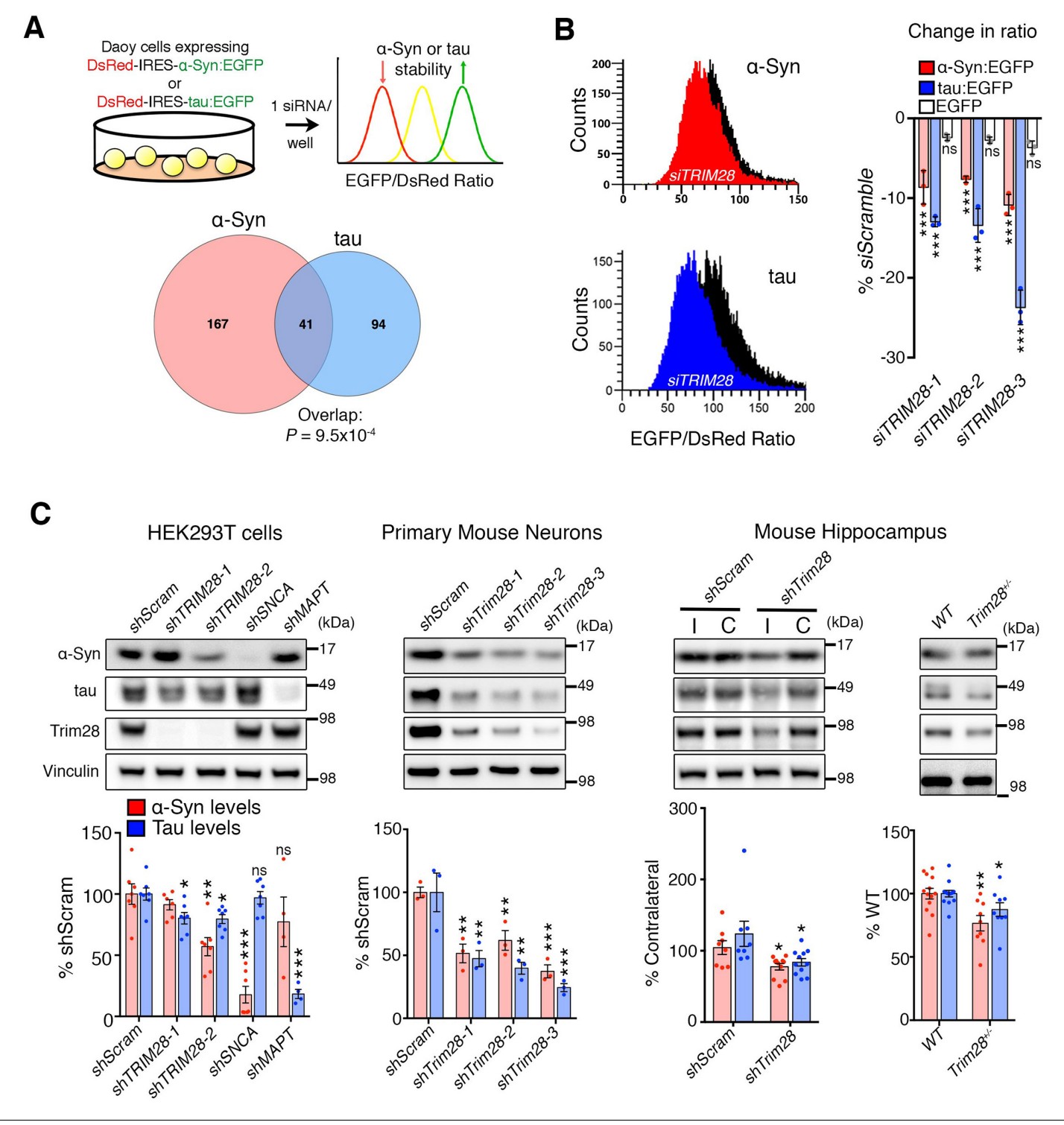

**Figure 1.** TRIM28 regulates levels of α-Syn and tau. (A) Schematic of screen approach (see also *Figure 1—figure supplement 1*). The ratio of either α-Syn:EGFP/DsRed or tau:EGFP/DsRed was measured using an arrayed siRNA library covering 2607 siRNAs in biological triplicates. Venn diagram shows significant overlapping hits from both screens. (B) Representative traces for α-Syn:EGFP/DsRed and tau:EGFP/DsRed ratios: the black curve represents a control condition (*siScramble*), while the red and blue curves represent *siTRIM28* in the α-Syn and tau cell lines, respectively. Quantified ratiometric scores for three independent siRNAs against *TRIM28* (*siTRIM28*-1,-2 and -3) is presented on the right. (C) Effects of different shRNAs targeting TRIM28 on endogenous α-Syn and tau in HEK293T cells (left panel, *shTRIM28-1* and *shTRIM28-2*), primary mouse neurons (middle panel, *shTrim28-1,1,-2,* and *-3*) and in adult mouse hippocampus (right panel, where 'I' denotes the injection side [ipsilateral] and 'C' denotes the uninjected side [contralateral, an

*Figure 1 continued on next page*

*Figure 1 continued*

internal control]). Rightmost panel depicts the effect of the loss of one allele of *Trim28* in the mouse brain. Data are presented as mean ± s.e.m. for each group. In **A**, p=*9.5 × 10⁻⁴*, hypergeometric test; in **B**, *n* = 3 per cell line, *** denotes p<*0.001*, One-Way ANOVA followed by Dunnet's multiple comparison test; in **c**, *n* = 4–13 condition, *, **, *** and ns denote p<0.05, p<0.01, p<0.001 and p>0.05, respectively, One-way ANOVA followed by Holm-Sidak post-hoc test in two leftmost panels and Student's t-test in two rightmost panels. Full statistical analyses for all figures are presented as ***Supplementary file 1.***

The following source data and figure supplements are available for figure 1:

**Source data 1.** List of modifier genes identified in convergent screens.

**Figure supplement 1.** Convergent screens targeting the steady state levels of α-Syn and tau identify common modifiers.

**Figure supplement 2.** Validation of shared modifiers between α-Syn and tau.

**Figure supplement 3.** Validation of *Trim28* knockdown and lack of effect on other neurodegenerative disease-causing genes.

accumulation. Therefore, convergent screens to find common modulators for α-Syn and tau levels would yield the most insight into these disease processes and possibly open up new avenues for therapeutic intervention. Importantly, targeting the root cause of the disease – protein accumulation – in an unbiased manner makes neither assumption about the mechanism of toxicity nor which cellular process is affected. Through convergent RNAi screens targeting the steady-state levels of α-Syn and tau, we found that TRIM28 regulates their levels and toxicity through their toxic nuclear accumulation.

## Results

### TRIM28 is a key regulator of α-Syn and tau levels

We employed a screening strategy similar to one recently used to identify key modulators of ATXN1 stability (*Park et al., 2013*; *Westbrook et al., 2008*), using high-throughput flow cytometry to monitor the steady-state levels of α-Syn and tau in a fluorescent bicistronic reporter system (*Figure 1A*). We ran parallel screens interrogating 2607 siRNAs targeting 869 potentially druggable – i.e. potentially can be targeted pharmacologically – genes to identify genes that modify the levels of both α-Syn and tau (*Figure 1—figure supplement 1A–C* and *Figure 1—source data 1*). Applying stringent criteria and validation steps to narrow down the list of putative modifiers of α-Syn and tau levels, we uncovered Tripartite motif-containing 28 (TRIM28) as the most robust common modulator (*Figure 1B*; *Figure 1—figure supplement 2* and *Figure 1—source data 1*). We confirmed this effect of *TRIM28* knock-down on α-Syn and tau stability using three independent siRNAs on our α-Syn and tau reporter cell lines along with a negative control cell line (DsRed-IRES-EGFP) to ensure that TRIM28 was not affecting the stability of EGFP (*Figure 1B*).

We next used species-specific shRNA lentiviral vectors to suppress *TRIM28* expression in human cells, in mouse primary cerebellar neurons, and in mouse hippocampus (*Figure 1C* and *Figure 1—figure supplement 3A,B*). In each case, partial reduction of TRIM28 led to a decrease of both α-Syn and tau, and virally reintroducing a human shRNA-resistant TRIM28 cDNA reversed this effect (*Figure 1—figure supplement 3C*). In addition, we found that deleting one allele of *Trim28* in mice led to a modest but significant decrease of α-Syn and tau levels in mouse brain (*Figure 1C*). Importantly, we observed little to no changes in *SNCA* nor *MAPT* transcript levels under conditions of TRIM28 downregulation (*Figure 1—figure supplement 3D*). Nor did reduction of TRIM28 affect other neurodegeneration-causing proteins we tested (*Figure 1—figure supplement 3E*). TRIM28 thus appears to be a key post-translational regulator of the steady state levels of α-Syn and tau.

### TRIM28 suppression blocks tau- and α-Syn-mediated toxicity in vivo

Based on studies showing that reducing disease protein levels can rescue neurodegeneration in other proteinopathy models (*Moreno et al., 2012*; *Park et al., 2013*), we tested whether decreasing

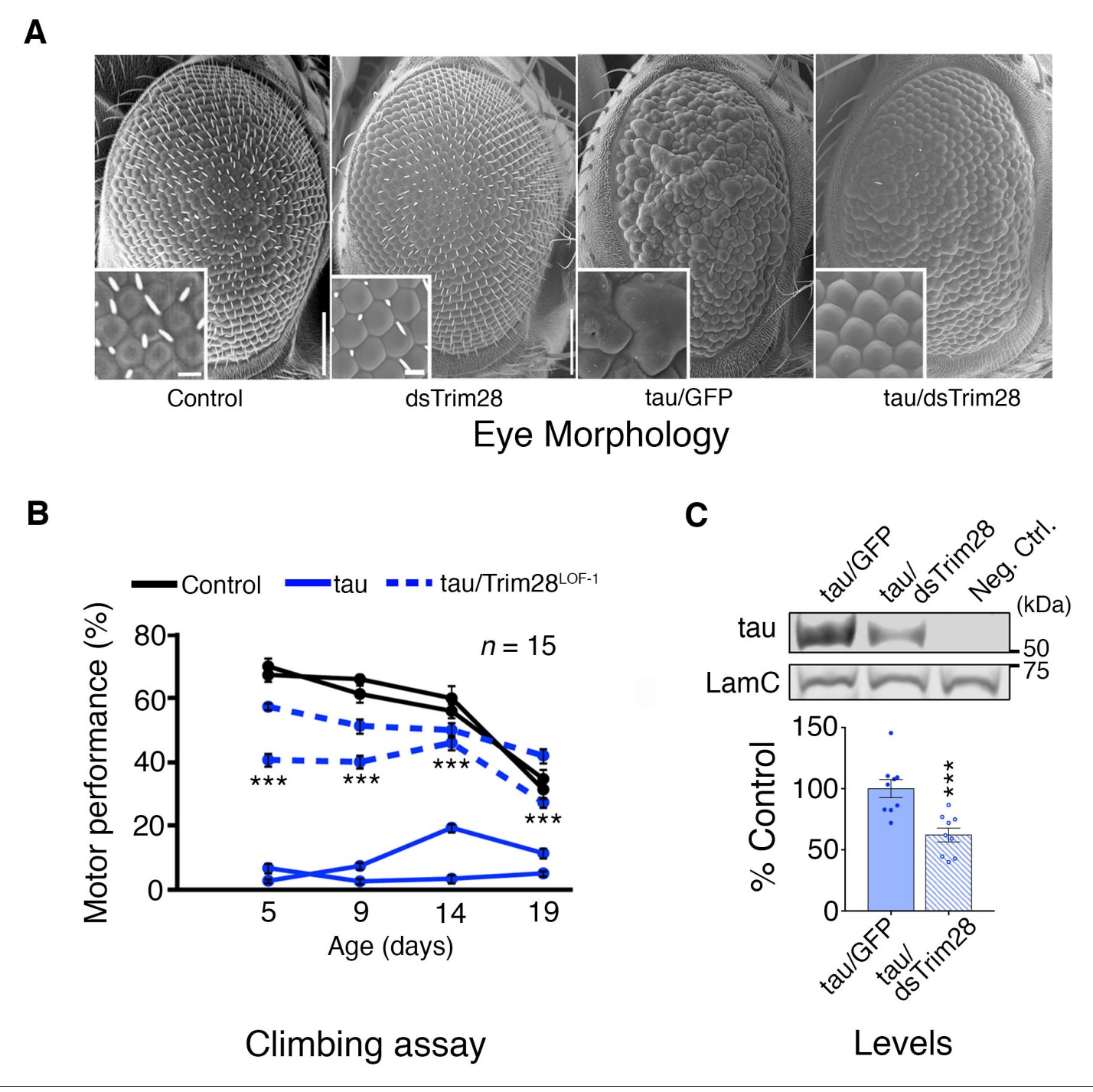

**Figure 2.** Loss of TRIM28 mitigates tau-mediated neurodegenerative phenotypes in *Drosophila*. (**A**) tau overexpression in the *Drosophila* eye produces a rough eye phenotype (third panel) compared to negative controls (first panel). Decreasing the levels of *Trim28* (dsTrim28) ameliorates this defect (fourth panel). Decreasing Trim28 alone does not result in any overt degenerative phenotypes (second panel). (**B**) Expression of tau in the *Drosophila* nervous system (solid blue lines) leads to motor performance deficits that can be quantified in a climbing assay when compared with control flies (black lines). This phenotype is mitigated by partial TRIM28 loss of function (hatched blue lines). Two independent cohorts (15 animals per replicate) are shown per genotype. (**C**) Western blot images and quantification showing decreased tau levels in the adult *Drosophila* retina, upon Trim28 knockdown. This reduction of tau protein levels is concordant with the suppression of tau phenotypes shown in A and B. Data are presented as mean ± s.e.m. for each group. In B, $n$ = 15 flies per replicate and 2 replicas per genotype *** denotes $p<0.001$, Two-Way ANOVA followed by Tukey-Kramer post-hoc test; in C, $n$ = 6 replicates per group, *** denotes $p<0.001$, Student's t-test. Scale bars in **A**: 100 μm (inset 10 μm).

*Figure 2 continued on next page*

*Figure 2 continued*

The following figure supplements are available for figure 2:

**Figure supplement 1.** Reduced function of TRIM28 alone does not produce abnormal behavioral phenotypes in *Drosophila* but rescues tau-mediated degeneration.

**Figure supplement 2.** TRIM28 loss does not inhibit Gal4 driver expression.

TRIM28 would mitigate the phenotypes resulting from overexpression of either α-Syn or tau. For a tauopathy model, we generated transgenic *Drosophila* that express wild-type human tau in the eye and develop a visible degenerative phenotype. Knockdown of the *Drosophila TRIM28* homolog (*bonus*) significantly reduced tau levels and mitigated eye degeneration (*Figure 2A*; *Figure 2—figure supplement 2C*). Similarly, expression of wild-type tau in the *Drosophila* nervous system led to motor deficits in the climbing assay, but two independent partial loss-of-function alleles of *TRIM28* improved this behavioral phenotype without reducing the transgene expression (*Figure 2B*; *Figure 2—figure supplement 1*; *Figure 2—figure supplement 2* and *Video 1*). To ensure that this effect was mediated through tau levels, we tested the effect of TRIM28 loss of function on transgenic tau levels and found indeed that reduction of TRIM28 could effectively decrease tau levels in this heterologous system (*Figure 2C*).

To test whether Trim28 mediates neurodegeneration in a synucleinopathy, we turned into a mouse model of α-Syn-overexpression-induced Parkinsonism (*Burré et al., 2012*), since the degenerative phenotypes of α-Syn-overexpressing flies are generally very mild. We co-injected lentiviral vectors expressing α-Syn together with a lentiviral vector expressing an shRNA against mouse *Trim28* into the *Substantia Nigra pars compacta*(SNc) of mice and evaluated dopaminergic cell integrity eight weeks afterward. Mice infected with viruses overexpressing α-Syn and a control shRNA suffered a 50% reduction in tyrosine hydroxylase (TH)-positive dopaminergic neurons (*Figure 3A,B*-**top panels** and *Figure 3—figure supplement 1C*). Remarkably, mice that received the *shTrim28* virus together with α-Syn overexpression maintained their dopaminergic cells in the SNc and the corresponding fibers in the striatum (*Figure 3A,B*-**bottom panels**). We confirmed that *Trim28* was expressed (and effectively knocked down in the case of *shTrim28* treatment) in these nigral neurons and that this rescue was due to a decrease in α-Syn levels and not the silencing of the α-Syn expression vector (measured by IRES-driven GFP expression, *Figure 3—figure supplement 1A,B*). Lastly, to test whether this phenotypic rescue would hold true in a non-acute paradigm, we tested whether loss of one copy of *Trim28* in mice could rescue some of the pathology observed in α-Syn-overexpressing mice (*Rockenstein et al., 2002*). These mice possess a transgene driving wildtype α-Syn under the control of the mThy1 promoter and demonstrate age-dependent accumulation and aggregation of α-Syn that is demarked by its S129 phosphorylated form. We found that a 50% reduction in Trim28 significantly reduced pathological, phosphorylated α-Syn accumulation in the hippocampus where it is highly expressed (*Figure 3C,D*) (*Chesselet et al., 2012*).

## TRIM28 stabilizes α-Syn and tau and drives their pathology

Since loss of TRIM28 rescued the neurodegenerative phenotypes, we surmised that increasing TRIM28 might exacerbate pathology. We performed bilateral lentiviral overexpression of TRIM28 in pre-symptomatic mouse models of

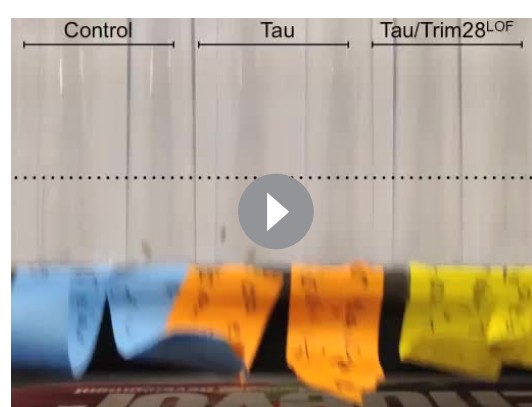

**Video 1.** Representative video. Partial loss of TRIM28 function in fruit flies overexpressing tau rescues motor behavior in the climbing assay. Video recording shows the three groups of animals tested in *Figure 2B*. The number of fruit flies climbing up 9 cm (dotted line) in 15 s was recorded.

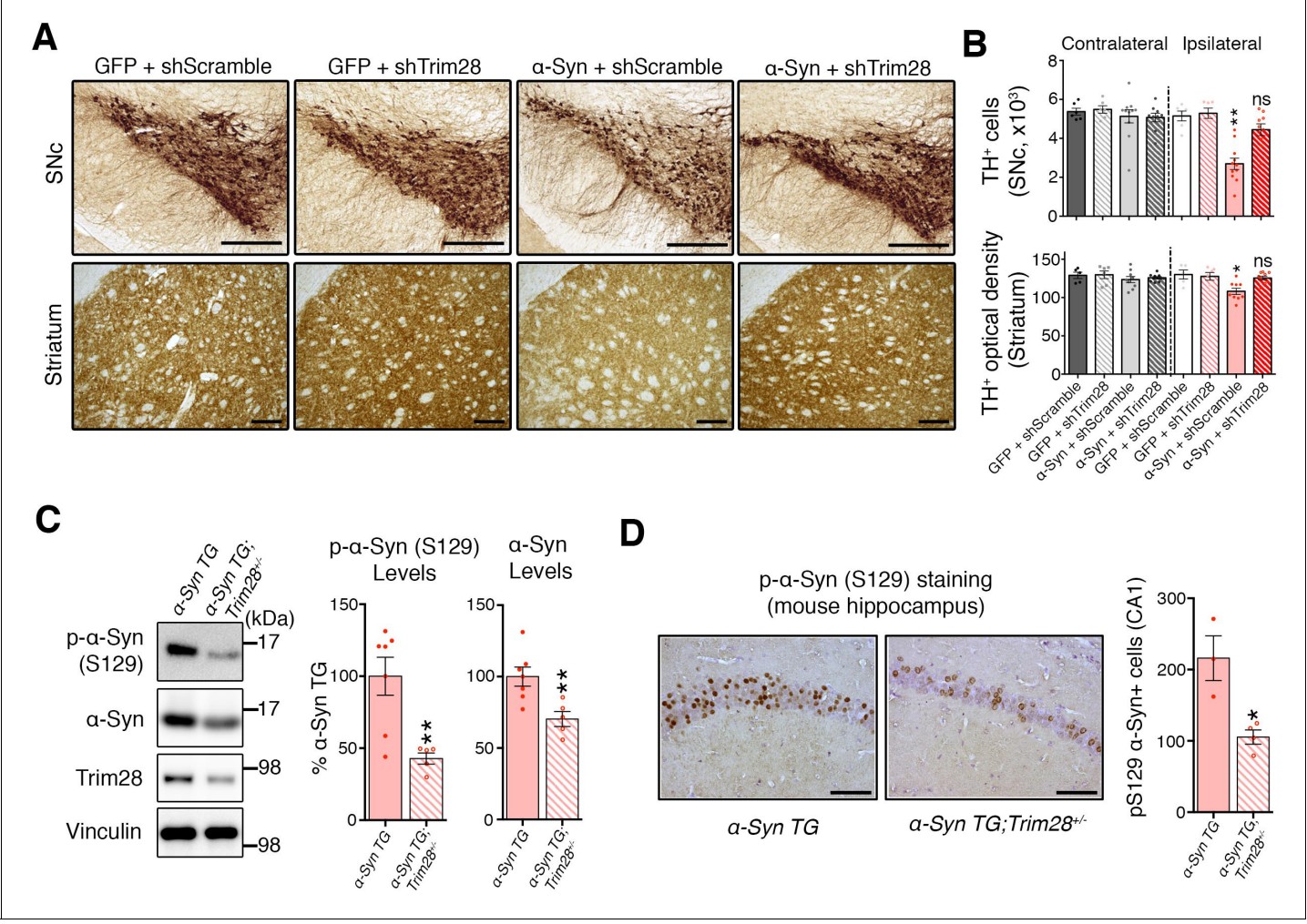

**Figure 3.** TRIM28 knockdown suppresses α-Syn-mediated neurodegenerative phenotypes in vivo. (A) Representative photomicrographs of midbrain sections stained for tyrosine hydroxylase (TH) at the level of the Substantia Nigra *pars compacta* (SNc, top panels) or in the striatum (bottom panels) on the ipsilateral side to the virus injection. (B) Stereological quantification of TH+ cells (top) in the SNc and quantification of optical density of TH+ fibers (bottom) is presented on the right. (C) Western blot analysis of 3.5 month old α-Syn transgenic (TG) mice carrying two (no mark, *Trim28+/+*) or one (*Trim28+/-*) copies of Trim28. (D) phosphorylation of α-Syn at serine 129 (pS129) staining at the level of the CA1 in these mice and quantification of positive cell numbers is presented on the right. Data are presented as mean + s.e.m. for each group. In B, *n* = 5–11 per group, * and ** denote p<0.05 and p<0.01, respectively, One-Way ANOVA followed by Holm-Sidak post-hoc test; in B–D, *n* = 3–7 mice per group, * and ** denote p<0.05 and p<0.01, respectively, Student's t-test. Scale bars in A: 400 μm (top panels) 150 μm (bottom panels), C: 50 μm.

The following figure supplement is available for figure 3:

**Figure supplement 1.** TRIM28 is expressed in the mouse SNc, and can be effectively knocked down in vivo.

synucleinopathy (*Rockenstein et al., 2002*) and tauopathy (*Yoshiyama et al., 2007*) and found that injection with TRIM28, but not control lentivirus, worsened each aspect of neuropathology tested in both models (*Figure 4A,B*; and *Figure 4—figure supplement 1A,B*). We were particularly intrigued to find pathological, phosphorylated, forms of α-Syn (S129) and tau (S396) accumulating at this earlier stage. Given the efficacy of TRIM28 at tightly regulating the levels of α-Syn and tau in a post-translational manner, we reasoned that TRIM28 was likely stabilizing these proteins therefore promoting their toxic accumulation. To test this, we generated doxycycline-inducible cell lines where we could carefully assess the half-lives of α-Syn and tau without altering global protein homeostasis (*Figure 5A*) (*Meerbrey et al., 2011*). Specifically, we cloned wild-type α-Syn and tau into doxycycline-inducible lentiviral vectors (pINDUCER system), infected SH-SY5Y cells and selected for clones

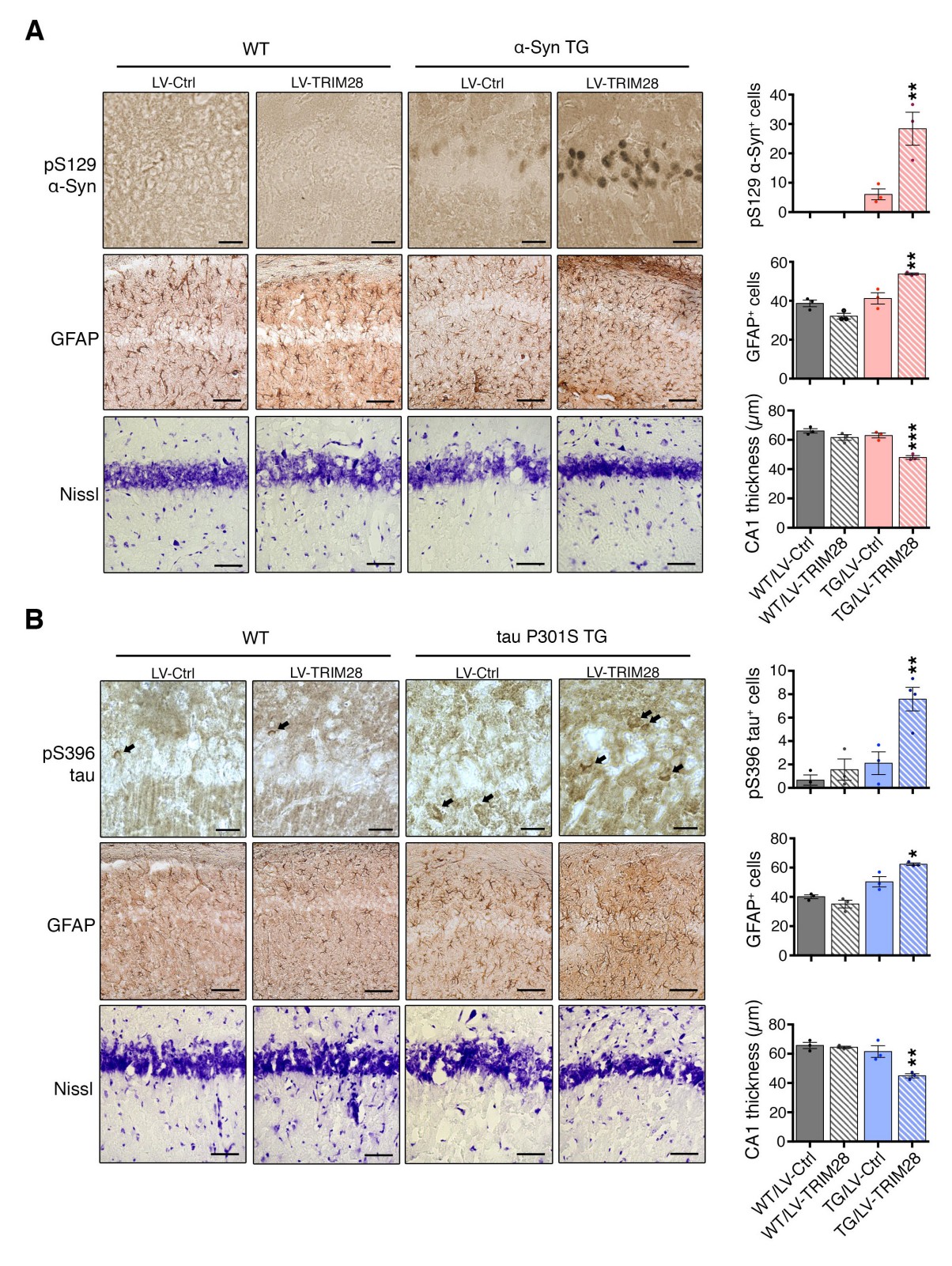

**Figure 4.** TRIM28 expression worsens histopathology in mouse models of synucleinopathy and tauopathy. Transgenic mice overexpressing α-Syn (**A**, mThy-Syn 'Line 61') or P301S tau (**B**, PS19) were injected at a presymptomatic stage in the hippocampus with lentiviruses expressing TRIM28. Pathological evaluation of phosphorylation of α-Syn at serine 129 (pS129, top panels); Glial Fibrillary Acidic Protein (GFAP, middle panels) as well as Nissl staining (bottom panels) in the CA1 region of α-Syn transgenic (TG) mice (solid bars) and their wild-type littermates (hatched bars) was performed
*Figure 4 continued on next page*

*Figure 4 continued*

(quantification on the right of each panel sets). Similar pathological evaluation of phosphorylation of tau at serine 396 (pS396 tau); GFAP; as well as Nissl staining in the CA1 region of P301S tau Transgenic mice. Data are represented as mean + s.e.m. In **A** and **B**, $n = 3$ for each genotype and treatment for each experiment, *, ** and *** denote $p<0.05$, $p<0.01$ and $p<0.001$, respectively, One-Way ANOVA followed by Holm-Sidak post-hoc test. Scale bars in **A** and **B**: 25 µm (top panels), 100 µm (middle panels) and 50 µm (bottom panels).

The following figure supplement is available for figure 4:

**Figure supplement 1.** Effective transduction of lentivirus expressing TRIM28 in the hippocampus intensifies biochemical accumulation of α-Syn and tau in prodromal mouse models of synucleinopathy and tauopathy.

that stably express either α-Syn or tau upon induction. We found that the half-lives of α-Syn and tau (measured following a 48 hr pulse of doxycycline) in this system resembled those previously observed by others (23.3 hr for α-Syn and 28.1 hr for tau [*Li, 2004*; *Min et al., 2010*]). Importantly, we found that TRIM28 lentiviral overexpression could significantly increase the half-life of each protein by approximately 50% (LV-TRIM28 vs. LV-Ctrl, *Figure 5B,C*), without affecting their RNA stability (*Figure 5—figure supplement 1*). Together, these studies suggest that TRIM28 promotes the accumulation of the amyloidogenic proteins α-Syn and tau.

## TRIM28 binds to and promotes the nuclear accumulation of α-Syn and tau

Given that TRIM28 regulates the stability of these two proteins, we next sought the consequence of TRIM28-mediated regulation of α-Syn and tau. We asked whether TRIM28 modulates α-Syn and tau by forming a complex with them and found that they did indeed interact, albeit weakly, in human cells (*Figure 6A*). We also used an alternative approach to confirm our findings: Bimolecular Fluorescence Complementation (BiFC; *Figure 6B*). We generated stable cell lines expressing N-terminal YFP tagged versions of α-Syn and tau (nYFP-α-Syn and nYFP-tau). By themselves, the cell lines did not generate any fluorescence (data not shown) but when a known interactor of each (in this case, tau) was co-transfected as a prey, we could elicit a strong fluorescent signal (tested by flow cytometry). We found that using TRIM28 as a prey could elicit a signal roughly eight times lower than that of tau thus confirming the transiency of the interaction. Intriguingly, TRIM28:α-Syn and TRIM28:tau complexes accumulated in nuclei after some time. To confirm that this effect was not an artifact of the BiFC approach, we overexpressed a mCherry-fused TRIM28 lentiviral construct in primary cerebellar granule neurons and looked at the endogenous localization of α-Syn and tau. Again, both by immunofluorescence and by biochemical fractionation, we found that viral expression of TRIM28 dramatically promoted the nuclear accumulation of these two proteins (*Figure 6C* and *Figure 6—figure supplement 1*). Given our previous data suggesting that the effects of TRIM28 on α-Syn and tau are posttranslational, we explored whether the RING domain of TRIM28 might play a role in its modulatory effect. To this end, we focused on the known E3-ligase catalytic activity of TRIM28. We overexpressed an inactive TRIM28 E3 ligase mutant ([C65A/C68A], 'TRIM28-Mut') (*Doyle et al., 2010*) and found that this mutant could no longer promote the toxic nuclear accumulation of these proteins (*Figure 6C* and *Figure 6—figure supplement 1*). Importantly, we found that the TRIM28 E3 ligase mutant, while less stable, could still bind α-Syn and tau (*Figure 6—figure supplement 2*), suggesting that its effect on α-Syn and tau nuclear accumulation is functionally, rather than structurally, driven. Taken together, TRIM28 governs the nuclear accumulation of α-Syn and tau through its E3-ligase domain, and its persistent expression aggravates the neuropathology of both proteins.

## Aberrant TRIM28 levels are linked to nuclear accumulation of α-Syn and tau in human synucleinopathies and tauopathies

If TRIM28 is indeed promoting accumulation of α-Syn and tau in a disease context, levels of TRIM28 could also be deregulated. We therefore obtained post-mortem tissue from individuals who had had PDD, AD, or progressive supranuclear palsy (PSP), along with age-matched controls, and examined the biochemical distribution of TRIM28. In all cases of synucleinopathy and tauopathy, a greater proportion of TRIM28 accumulated in an insoluble form (Formic Acid fraction) than the more soluble form (RIPA fraction) (*Figure 7—figure supplement 1A–D*). Subcellular fractionation on these post-

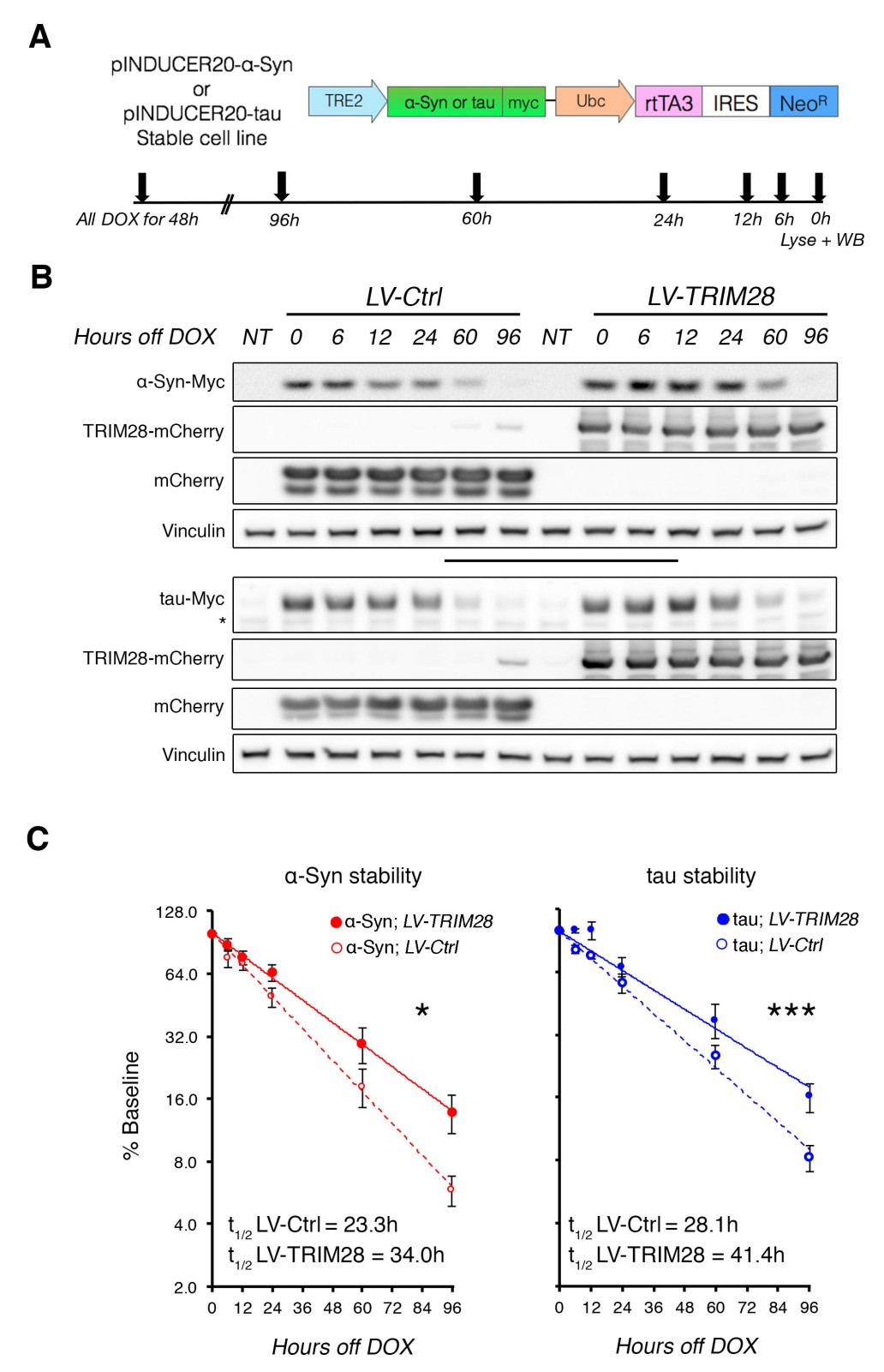

**Figure 5.** TRIM28 stabilizes α-Syn and tau protein levels. (**A**) Schematic of viral vector used to generate stable cell lines (adapted from [**Meerbrey et al., 2011**]) and assay design. (**B**) Representative western blots of transgenic α-Syn (top) and tau (bottom) cell lines following different times of doxycycline (DOX) removal; NT denotes non-DOX treated cells; * denotes detection of endogenous tau (not quantified). (**C**) Quantification of α-Syn and tau stability are presented as a % of baseline and fit to a logarithmic (log₂) scale. Dashed lines denote control groups whereas solid lines
*Figure 5 continued on next page*

*Figure 5 continued*

denote TRIM28 overexpression. Data are presented as mean ± s.e.m. for each group. In **C**, *n* = 6 per group, per time point. Protein levels follow a one-phase exponential decay with both curves being significantly different from one another (Comparison of fits, p=0.0111, α-Syn;*LV-Ctrl* vs. α-Syn;*LV-TRIM28*; p=0.0003, tau; *LV-Ctrl* vs. tau;*LV-TRIM28*).

The following figure supplement is available for figure 5:

**Figure supplement 1.** TRIM28 overexpression does not affect SNCA and MAPT RNA stability.

mortem tissues revealed that α-Syn and tau accumulated in the nucleus in most cases of synucleinopathy and tauopathy (*Figure 7—figure supplement 2A,B*). Then, we looked at α-Syn and tau subcellular localization in relation to TRIM28 in the context of PD and AD, respectively. We found that α-Syn and tau nuclear co-localization with TRIM28 was greater in PD and AD than in age-matched controls (*Figure 7A,B*). Taken together, our findings suggest that TRIM28 is aberrantly increased in cases of synucleinopathy and tauopathy and is pathologically associated with the nuclear accumulation of α-Syn and tau in a diseased state.

## Discussion

The mechanism(s) underlying the neurotoxicity of α-Syn and tau have been difficult to pin down (*Cookson and van der Brug, 2008*; *Ward et al., 2012*). Although accumulating evidence suggests that α-Syn and tau pathology propagate from cell to cell, the exact mechanism driving their initial accumulation and toxicity remains unclear. Given that both α-Syn and tau accumulate in Parkinson Disease Dementia and Lewy Body Dementia (*Moussaud et al., 2014*; *Sengupta et al., 2015*), we rationalized that there must be some proteins that regulate both of them. Thus, when screening for posttranslational regulators that modulate the levels of each protein, we were most interested in regulators that can modify the levels of both α-Syn and tau. Through such convergent screens, we found that TRIM28 regulates the levels of both disease-driving proteins. Our finding that TRIM28 regulates α-Syn and tau by promoting their nuclear localization and accumulation dovetails nicely with studies indicating that altered cellular distribution of α-Syn and tau contributes to neurodegeneration (*Fares et al., 2014*; *Fernández-Nogales et al., 2014*; *Kontopoulos et al., 2006*). Indeed, several lines of evidence suggest that mislocalization or missorting of α-Syn and tau as a result of mutations, post-translational modification or overexpression, rather than their respective aggregation in the form of Lewy bodies and NFTs, are what promotes neurodegeneration (*Frandemiche et al., 2014*; *Wilson et al., 2004*). For instance, α-Syn is much more toxic in a fly model of synucleinopathy when it is tagged with a nuclear localization sequence (NLS) (*Kontopoulos et al., 2006*). Further, PD-causing mutations in α-Syn increase its nuclear accumulation (*Fares et al., 2014*; *Kontopoulos et al., 2006*). Moreover, postmortem studies of tissue from AD and Huntington disease patients indicate that misfolded tau accumulates in the nucleus of neurons in the form of rod-like deposits (*Fernández-Nogales et al., 2014*; *Vuono et al., 2015*). Our finding that TRIM28 drives the nuclear accumulation of α-Syn and tau provides new mechanistic insight into the steps that lead to the pathogenicity of these proteins and highlights a shared pathway for therapeutic targeting.

The molecular mechanism through which TRIM28 stabilizes and mediates the nuclear localization of α-Syn and tau is not elucidated at this point. Indeed, it is possible that TRIM28 mediates its effects on α-Syn and tau indirectly through a yet undiscovered mediator, through epigenetic remodeling or indirect binding and/or modification of an intermediate (*Cheng, 2014*). However, the evidence we present suggests that this less likely of a mechanism. For instance, we show that TRIM28 does not affect the RNA of α-Syn and tau but rather their protein stability. Moreover, we show that TRIM28 can form a complex with the two proteins and that its effect on nuclear accumulation is mediated via its E3-ligase domain. This would therefore suggest that TRIM28 may mediate this effect through SUMOylation or ubiquitination. Since SUMOylation has previously been suggested to mediate the nuclear localization of certain proteins, we hypothesize that TRIM28 acting as a SUMO ligase for α-Syn and tau. Future studies will dissect such a mechanism through the use of genetically engineered mice that permit monitoring of either modification in vivo.

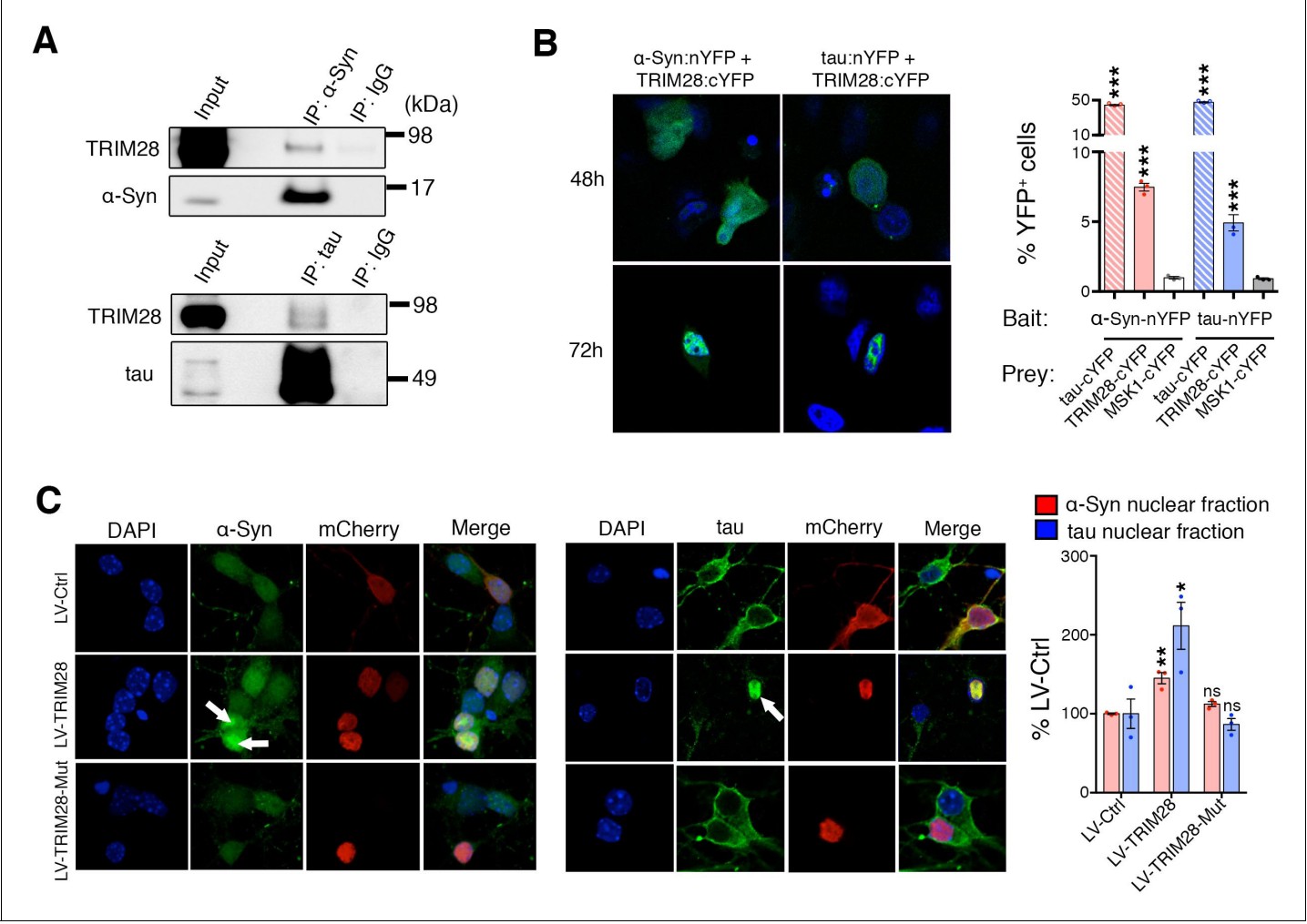

**Figure 6.** TRIM28 binds to and drives the nuclear localization of α-Syn and tau. (**A**) Immunoprecipitation for endogenous α-Syn (top panel) and tau (bottom panel) from HEK293T cells showing the interaction between the former proteins and TRIM28. (**B**) Bimolecular Fluorescence Complementation studies using stable cell lines expressing either α-Syn or tau fused to an n-Terminal moiety of YFP (α-Syn:nYFP or tau:nYFP, respectively) and TRIM28 (solid), tau (positive control, hatched) or MSK1 (negative control, white) fused to a c-terminal moiety of YFP (TRIM28:cYFP, tau:cYFP and MSK1:cYFP, respectively). Visualization of the epifluorescence generated through protein-protein interaction is depicted in the photomicrographs and quantified by flow cytometry on the right. Note the nuclear accumulation of the interacting proteins 72 hr following transfection. (**C**) Primary mouse neurons infected with lentiviruses harboring TRIM28, TRIM28 E3 ligase mutant and control were stained for endogenous α-Syn (left panel), and tau in (middle panel) and the relative nuclear fraction of each was quantified (right panel). Arrows point to endogenous α-Syn and tau in the nucleus. Data are represented as mean + s.e.m. In B, n = 3 per group and *** denotes p<0.001, One-Way ANOVA followed by Holm-Sidak post-hoc test; in C, n = 3 per group and *, ** and ns denote p<0.05, p<0.01 and p>0.05, respectively, One-Way ANOVA followed by Holm-Sidak post-hoc test.

The following figure supplements are available for figure 6:

**Figure supplement 1.** TRIM28 promotes the nuclear localization and accumulation of α-Syn and tau.

**Figure supplement 2.** TRIM28 catalytic mutant retains some binding capacity to α-Syn and tau.

In the present study, we found that TRIM28 has important post-transcriptional functions in the brain. While the *Trim28* null mice are embryonic lethal (*Cammas et al., 2000*), we found that the *Trim28⁺/⁻* mice looked and behaved normally and that 50% loss of Trim28 was sufficient to reduce α-Syn and tau and ameliorate toxicity. This finding is particularly encouraging when it comes to therapeutics as it suggests that only partially inhibiting TRIM28 function may be effective at blocking neurotoxicity. Moreover, having discovered that mutating the E3 ligase catalytic activity of TRIM28

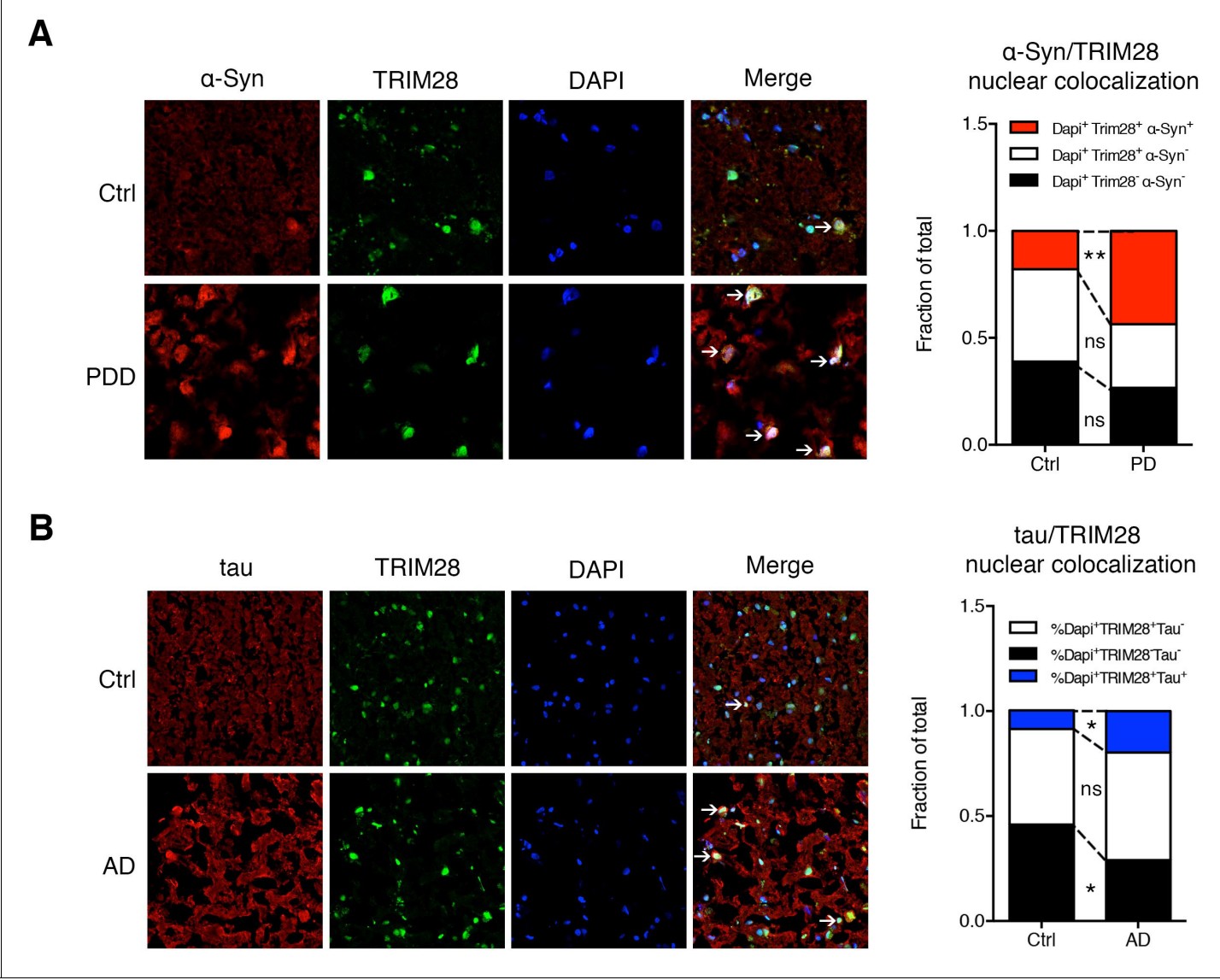

**Figure 7.** α-Syn and tau colocalize with TRIM28 in the nucleus in cases of synucleinopathy and tauopathy. (A) Representative photomicrographs of the medial frontal gyrus of cases of PDD and age-matched controls stained for α-Syn, TRIM28 and DAPI. The relative proportion of DAPI positive cells were quantified and are presented as the fraction of total nuclei counted. (B) Representative photomicrographs as in A but for cases of AD and respective age-matched controls, quantification on the right. Data are represented as a fraction of total. In A, $n = 4$ per post-mortem group, ** and ns denote $p<0.01$ and $p>0.05$, respectively, Student's t-test; in B, $n = 5$ per post-mortem group, * and ns denote $p<0.05$ and $p>0.05$, respectively, Student's t-test.

The following figure supplements are available for figure 7:

**Figure supplement 1.** TRIM28 levels are deregulated in human cases of synucleinopathy and tauopathy.

**Figure supplement 2.** α-Syn and tau accumulate in the nucleus in cases of synucleinopathy and tauopathy.

effectively is sufficient to regulate α-Syn and tau levels and for this specific effect behaves like a null allele, narrows down a target domain within TRIM28 for inhibition. This finding sets the stage for future studies to dissect the native function of the E3 ligase domain of TRIM28 versus its other essential domains and provides a potential opportunity for targeting TRIM28 as a source of therapy for these debilitating disorders.

The unusual accumulation of TRIM28 in cases of synucleinopathy and tauopathy was a curious finding, though it remains solely an observation of correlative nature at this point. While we found that both α-Syn and tau co-localized with TRIM28 in the nucleus in cases of PD and AD, respectively, it was also interesting to find that TRIM28 accumulated in the insoluble biochemical fractions from post-mortem tissue. Moreover, though TRIM28 accumulation occurs in both diseases, irrespective of the driving pathological protein, α-Syn and tau appeared to more selectively accumulate in the nucleus in the context of their respective disease. Teasing out the precise mechanisms that affect TRIM28 accumulation as well as α-Syn and tau nuclear localization in human tissue will provide additional context for the understanding of disease pathogenesis.

We posit two potential mechanisms through which deregulated TRIM28 activity could promote neurodegeneration. By driving α-Syn and tau to the nucleus, TRIM28 could: (1) prevent their degradation by native quality control mechanisms, thereby enhancing their overall bioavailability and toxicity (a passive mechanism); or (2) allow for a toxic gain of function in the nucleus (an active mechanism) (model, *Figure 7—figure supplement 2*). Though some studies have suggested various roles for α-Syn and tau in the nucleus (*Fernández-Nogales et al., 2014*; *Kontopoulos et al., 2006*; *Sultan et al., 2011*), careful studies testing the kinetics and the outcome of this nuclear build-up will surely shed light on disease pathogenesis.

As the sensitivity of the brain to protein levels becomes clearer, and more neurodegenerative diseases are found to involve elevated levels of more than one protein (*Moussaud et al., 2014*; *Ramanan and Saykin, 2013*), it will become more important to identify shared modifiers and regulatory mechanisms of the steady-state levels of disease proteins, both to understand pathogenesis and to find the best candidate targets for therapeutic interventions.

# Materials and methods

## Abbreviations

α-Syn, alpha-synuclein; AD, Alzheimer disease; BCA, bicinchoninic acid; BiFC, bimolecular fluorescence complementation; CA1, Cornu Ammonis 1; 'C' or CONTRA, contralateral; cDNA, complementary DNA; DsRed, Discosoma red fluorescent protein; DMEM, dulbecco's modified eagle medium; DOX, Doxycycline; EGFP, enhanced green fluorescent protein; FACS, fluorescent activated cell sorting; FBS, fetal bovine serum; FTD, Frontal Temporal Dementia; GFAP, Glial Fibrillary Acidic Protein; HEK, human embryonic kidney; HTS, high throughput sampler; 'I' or IPSI, ipsilateral; IP, immunoprecipitation; IRES, internal ribosomal entry site; LBD, Lewy Body Demetia; LV, Lentivirus; Mut, mutant; PD, Parkinson disease; PDD, Parkinson disease with dementia; PFA, paraformaldehyde; RNAi, ribonucleic acid interference; PSP, Progressive Supranuclear Palsy; RIPA, Radioimmunoprecipitation assay (buffer); RRE, Rev response element; siRNA, small interfering ribonucleic acid; shRNA, short hairpin ribonucleic acid; SNc, substantia nigra *pars compacta*; SUMO, Small Ubiquitin-Like Modifier; TH, tyrosine hydroxylase; TU, Transducing units; TRIM28, Tripartite Motif Containing 28; UBC9, Ubiquitin carrier protein 9; and VSVG, vesicular stomatitis virus-G.

## Antibodies, shRNAs, plasmids and primers

The collection of antibodies (and their concentrations), shRNAs (and their sequences), Plasmids (and their construction) as well as qRT-PCR primers used are presented in *Supplementary file 2*.

## Virus generation, concentration and infection

Lentiviral vectors (pHAGE, L302, LV-mCherry, pGIPz or pZip) and their respective packaging vectors (VSVG, Rev and RRE for L302; psPAX2 and pMD2G for LV-mCherry, pGIPz and pZip) were cotransfected into HEK293T cells in a 1:1:1:1 or 4:3:1 molar ratio, respectively. Media was changed 16 hr following transfection to low volume media (5 mL for a 10 cm dish). Media was collected at 48 hr following transfection, replaced with fresh media (5 mL) and collected again at 72 hr. Viral supernatant was cleared from cell debris via centrifugation (10 min at 4000 rpm) as well as filtration through a 0.45 µM polyethersulfone membrane (VWR). Cleared supernatants were concentrated using Lenti-X concentrator (Clontec) to 1/50–1/100 of the original volume. Viruses were titered using the Open Biosystems (Thermoscientific, Waltham, MA) method (counting either fluorescence-positive colonies or puromycin-resistant colonies).

## Generation of stable cell lines

DsRed-IRES-α-Syn:EGFP, DsRed-IRES-tau:EGFP, DsRed-IRES-EGFP, α-Syn:nYFP and tau:nYFP cell lines were generated as previously described (*Park et al., 2013*). Briefly, each construct was cloned into a pHAGE vector, packaged into lentiviruses and infected into Daoy cells at low multiplicity of infection (0.3) to promote single copy integration. Cells were then selected by puromycin (1 µg/mL) and single clonal populations were obtained using fluorescence activated cell sorting (FACS) using an Aria II cell sorter (BD Biosciences) and sorting a single double-positive (DsRed and EGFP) cell per well. Clones were expanded and the ones with the best qualities for screening (i.e., low transgene expression, low variation, homogenous population) were expanded for screening purposes.

## Flow cytometry analysis and cell-based RNAi screen

Flow cytometry was carried out on a LSRII Fortessa coupled with an HTS module (BD Biosciences). Gates for analysis were set using single-positive and double-positive cells as well as negative cells. For the RNAi Screens, DsRed-IRES-α-Syn:GFP or DsRed-IRES-tau:EGFP cells were plated onto 96-well round bottom plates. For each RNAi plate, cells were plated in triplicate to assay for technical variance. The following day, individual siRNAs were transfected as previously reported (*Park et al., 2013*) and incubated for 72 hr for downstream flow cytometry analysis. For each plate tested, Z-Scores were calculated by comparing each sample to the plate mean. A cut-off Z-Score of 1.5 was set in either direction for a first pass of hits. Next, individual t-tests were performed on hits by comparing them to the average of three scrambled siRNA sequences (with Low-, Medium-, and High GC content) to control for variability between triplicates. Hits meeting a p-value cut-off of <0.05 were kept for downstream analysis. Top hits from each screen met the Z-score and p-value cut-off and had more than two siRNAs with a significant effect (regardless of the directionality of said effect) (*Birmingham et al., 2009*). Validation of top hits was performed by ordering new siRNAs against most robust hits and revalidating in the same platform but this time including a DsRed-IRES-GFP cell line as a negative control (see *Figure 1—figure supplement 2* and *Figure 1—source data 1*). Hits that significantly altered the DsRed-IRES-GFP cell line in the same direction were excluded. Further downstream analysis consisted of generating viral shRNAs against the top hits (using the human pGIPz collection, ThermoScientific) and testing for their effect on endogenous α-Syn and tau levels in HEK293T cells as shown in *Figure 1C*.

## Cell culture, lysate preparation and western blot analysis

Daoy, HeLa or HEK293T cells (obtained and certified from ATCC) were cultured in DMEM (Invitrogen) containing 10% FBS and antibiotics (Penicillin/Streptomycin). These lines are not misidentified as per ICLAC and are free from mycoplasma contamination. siRNAs were transfected using DharmaFECT (Dharmacon) and incubated for 72 hr prior to analysis by flow cytometry or western blot. shRNA viruses were generated as described below, infected into cells, and left for 9 days prior to analysis. Plasmids were transfected using Lipofectamine 2000 (Invitrogen) and left to express from 30 to 72 hr, depending on the downstream application. Unless otherwise mentioned, cells were lysed with RIPA buffer (50 mM Tris-HCl [pH 8.0], 150 mM NaCl, 1 % NP-40, 0.5% sodium deoxycholate, 0.1% SDS) supplemented with protease inhibitors (Roche) on ice for 20 min with vortexing. Lysates were cleared by centrifugation (20 min, 15,000 r.p.m, 4°C) followed by protein quantification via BCA assay (Pierce). Sample buffer with reducing agent was added to each lysate followed by a 10 min incubation at 95°C. Samples were spun down and run on a 4–12% Bis-Tris gel, transferred to a nitrocellulose membrane and blocked for one hour with 5% non-fat milk prior to primary antibody incubation.

## Primary neuron culture

Primary cerebellar granule precursors were generated as previously described (*Aleyasin et al., 2007*) with slight modifications. Briefly, cerebella from P5-P9 mice were dissected and dissociated with trypsin. Granule neurons were specifically isolated using a Percoll gradient following which single cell suspensions were plated onto 12-well plates previously coated with poly-L-lysine. Cells were left to recover for 24 hr prior to infection with 4 µL of concentrated virus (~1E7 Transducing units [TU] / µL). Cerebellar granule neurons were used as opposed to cortical and hippocampal cultures

due to their ease in infectivity (>80% infected, data not shown) and their homogeneity thus allowing for clean biochemical experiments.

## Stereotaxic injections

Stereotaxic injections were performed as previously described (*Lasagna-Reeves et al., 2015b*). Briefly, 8–12 week-old mice were deeply anesthetized with a ketamine/xylazine cocktail; their heads were shaven and cleaned using aseptic solution. A midline incision was performed to reveal Bregma and a burr hole was drilled at the appropriate coordinates. Mice were injected with concentrated lentiviral solution (1E7 TU / µL; 1 µL unilateral, left hemisphere for SNc injections, 3 µL of 2.5E7 TU / µL bilaterally for hippocampal injections) at a flow rate of 0.125 µL/min. For the hippocampal injections, coordinates relative to Bregma were: 2.1 mm posterior, 1.3 mm lateral and 1.5 mm ventral. For SNc injections, coordinates relative to Bregma were 3.2 mm posterior, 1 mm lateral and 4.2 mm ventral. Incisions were glued together using VetBond (3 M). Mice were followed for proper recovery following injection and received daily fluids (0.5 mL subcutaneous saline) and analgesics (100 µL IP Ketoprofen) for 72 hr following the injection.

## Immunofluorescence staining

Free floating or slide-mounted sections were stained as before (*Lasagna-Reeves et al., 2015a*; *Rousseaux et al., 2012*). Briefly, sections were permeabilized and blocked in PBS containing 0.3% Triton X-100 and 5% normal goat serum. Samples were incubated with primary antibody overnight in blocking buffer and were washed three times the next day prior to adding fluorescent-conjugated secondary antibody (either Alexa 488 or Alexa 568, 1:700 concentration) and incubating an additional hour at room temperature. Tissue was then washed three more times during and DAPI was added during one of the washes to permit nuclear visualization. Slides were coverslipped using fluoromount and imaged using a Zeiss LSM710 confocal microscope.

## Mouse breeding and colony management

mThy1-α-Syn (Line 61 [*Rockenstein et al., 2002*]) mice were a gift from Marie-Francoise Chesselet and were bred and maintained on a C57Bl/6J background. Note that in our hands, these mice exhibited stronger pathological phenotypes than those originally reported; this may be due to their background (data not shown). tau transgenic mice, overexpressing human tau with a P301S mutation (PS19 [*Yoshiyama et al., 2007*]), were obtained from the Jackson Laboratory were maintained on a C57Bl/6J background. *Trim28*$^{+/-}$ mice were generated by crossing *Trim28*$^{flox/+}$(*Cammas et al., 2000*) (B6.129S2[SJL]-*Trim28*$^{tm1.1lpc}$/J, Jackson Laboratory) mice with CMV-Cre mice (*Schwenk et al., 1995*) (B6.C-Tg[CMV-cre]1Cgn/J, Jackson Laboratory). Genotyping for *Trim28*$^{+/-}$ mice was performed using standard methods and the following primers to detect the recombined allele: 5′-TTGTTTATTTGGGAATGGTTGTTC-3′ and 5′-GCGAGCACGAATCAAGGTC-3′. *Mapt*$^{-/-}$ mice were a generous gift from J. Noebels (BCM) and *Snca*$^{-/-}$ mice were obtained from the Jackson Laboratory (B6;129 X 1-*Snca*$^{tm1Rosl}$/J). C57Bl/6J mice used for stereotaxic surgery in this study were aged 8–12 weeks. For all studies, mice of both sexes were used, unless specified. Up to five mice were housed per cage and kept on a 12 hr light; 12 hr dark cycle and were given water and standard rodent chow *ad libitum*. All procedures carried out in mice were approved by the Institutional Animal Care and Use Committee for Baylor College of Medicine and Affiliates.

## Stability assay

SH-SY5Y cell lines stably expressing α-Syn or tau were generated using the pINDUCER system (*Meerbrey et al., 2011*). Briefly, myc-tagged α-Syn or tau was inserted into the pINDUCER20 (ORF-UN) cassette. Lentivirus was generated as above and naïve SH-SY5Y cells were infected and selected for more than a week in geneticin-containing medium (G418, 150 µg/mL). Cells were then split into a 24 well plate and treated with doxycycline (DOX, 100 ng/mL) for 48 hr. DOX-containing media was replaced with regular media (containing no tetracycline) at indicated time points. All cells were harvested at the same time as described above for downstream western-blot testing. Half-lives were calculated from normalized densitometric values obtained by Image J and solving x when y = 50% for the following equation: $y = f(e^{-kx})$ as previously described (*Li, 2004*).

Alternatively, RNA was isolated from the indicated time points qPCR was performed using the primers presented in *Supplementary file 2*.

## RNA extraction and quantitative real-time PCR (qPCR)

HEK293T cells were infected with lentiviral constructs harboring shRNAs to *TRIM28* or scrambled controls. Eight days following infection and puromycin selection, cells were spun down and RNA was extracted using the miRNeasy kit (Qiagen) according to the manufacturer's instructions. For the in vivo quantification of genes expression RNA was extracted from whole mouse brain tissue.

RNA was quantified using the NanoDrop 1000 (Thermo Fisher) and quality assessed by gel electrophoresis. cDNA was synthesized using a Quantitect Reverse Transcription kit (Qiagen) starting from 1 µg of RNA. Quantitative RT-polymerase chain reaction (qRT-PCR) experiments were performed using the CFX96 Touch Real-Time PCR Detection System (Bio-Rad Laboratories) with PerfeCTa SYBR Green FastMix, ROX (Quanta Biosciences). Real-time PCR results were analyzed using the comparative Ct method and normalized against the housekeeping gene *Hs-GAPDH* or *mm-Hprt*. The range of expression levels was determined by calculating the standard deviation of the ΔCt (*Pfaffl, 2001*). Primers used to amplify specific exons of the target genes have been designed across introns to distinguish spliced cDNA from genomic contamination. Primers sequences are presented in *Supplementary file 2.*

## *Drosophila* methods

Drosophila lines were obtained from the Bloomington Stock Center (Indiana) and VDRC (Vienna). Human tau transgenic flies were generated by cloning human wild-type 4 repeat tau under the control of a UAS promoter, and injected in w- embryos following standard transgenesis procedures. For eye assays, expression of wild-type 4 repeat human tau was driven to the eye using GMR-Gal4 and fruit flies, cultured at 28°C, were processed for scanning electron microscopy analysis as previously described (*Park et al., 2013*). tau levels were analyzed in *Drosophila* expressing tau in the adult eye under the control of Rh1-Gal4 at 28.5°C. Animals were aged for ten days, heads were collected, homogenized in LDS Buffer (Invitrogen), 10% β-mercaptoethanol and loaded in 4–12% Bis-Tris gel, transferred to a nitrocellulose membrane, blocked for one hour with 5% non-fat milk and incubated overnight with primary antibody. For motor performance analysis, expression of tau was driven specifically in neurons with elav-Gal4, and crosses were performed at 26°C. Motor performance was assessed in a climbing assay as previously described (*Park et al., 2013*). Briefly, % motor performance was calculated as the percentage of flies that managed to climb up 9 cm in 15 s (averaged over 10 trials per time point).

For a list of all *Drosophila* strains used in this study, please see *Supplementary file 2*, tab 'Fly lines'. Trim28[LOF-1] corresponds to the allele P{EPg}bonHP32434, which is a mobile activating P element inserted in the 5'UTR of bonus (the Drosophila homologue of Trim28). These P elements can cause loss of function when inserted in the orientation of the negative strand of the inserted gene. Trim28[LOF-2] carries a loss of function allele [21B], previously characterized as missing all but 12 bp of exon 1 and the first 324 bp of intron 1. We confirmed that Trim28[LOF-1] is a loss of function allele of bonus because it failed to complement Trim28[LOF-2], which has already been described as a bonus LOF allele (*Beckstead et al., 2001*).

## Quantification of dopaminergic cell survival

Eight weeks after virus injection, mice were deeply anesthetized and transcardially perfused with 0.9% saline followed by 4% PFA. Brains were removed, placed in 4% PFA overnight for post-fixation and dehydrated using 10% Sucrose in PBS (twice a day for 3 days). Brains were frozen in O.C.T. compound (Tissue-Tek) and 30 µm free-floating sections were cut from the striatum and Substantia Nigra *pars compacta* (SNc). Tyrosine hydroxylase and Nissl staining in both the SNc and Striatum was performed as previously described (*Rousseaux et al., 2012*). Unbiased stereology using the optical fractionation method was used to estimate total dopaminergic cell counts in the SNc of each treatment group (StereoInvestigator 11, MBF Bioscience). Samples missing multiple sections or with staining artifacts were excluded from stereological analysis prior to unblinding. For striatal TH optical density quantification, images were captured using a 10x objective (both Ipsi- and Contralateral sides). Pictures were processes using an in house automated Adobe Photoshop CS5 workflow (Automator,

Mac OSX) to convert the image to gray scale, invert (so that the intensity of staining is a positive value) and saved as a new file. Five regions were sampled per picture in the TH-positive striatum and these were normalized to negative stained background (corpus callosum). Normalized values across three independent sections were used to compare the relative TH optic density between groups.

## Pathology assessment in mouse models of synucleinopathy and tauopathy

An experimenter blind to the treatment or mouse genotype injected 6–8 week old α-Syn and tau transgenic mice (mThy1-α-Syn and P301S tau) and their respective wild-type littermate controls bilaterally with virus harboring TRIM28-mCherry or mCherry as a control into the CA1 region of the hippocampus (*Dhungel et al., 2015*). One month following injection, mice were euthanized by isofluorane inhalation followed by cervical dislocation and decapitation. Brains were dissected, split down the midline and either post-fixed for 48 hr in 4% PFA or collected for downstream biochemical analysis. Post-fixed brains were dehydrated in 10% sucrose (as above) and sectioned directly on slides at a thickness of 20 μm. Alternatively, sections were paraffin embedded and sectioned at 5 μm for IHC analysis. Slides were then stained for pS129 α-Syn, pS396 tau and GFAP and visualization was performed using the ABC detection method (Vectastain, Vector Labs) as previously described (*Lasagna-Reeves et al., 2015b*). Nissl staining was performed as described previously (*Rousseaux et al., 2012*). Quantification of each pathological parameter was performed at an area surrounding the injection site (though not directly beside the needle tract to avoid staining artifacts) on three independent sections per animal with at least three animals for each tested condition. Counts were performed in a defined area (i.e. 1000 × 1000 pixels) using ImageJ 1.47v.

## Immunoprecipitation

Immunoprecipitation of protein complexes was performed as previously described (*Burré et al., 2010*). Briefly, cell lysis was performed on ice for 20 min with brief vortexing using lysis buffer (1% Triton X-100, 150 mM NaCl, 10 mM Tris pH 8.0, 10% glycerol, 20 mM N-ethyl maleimide and protease inhibitors [Roche]). Cell debris were removed by centrifugation (20 min at 15,000 r.p.m, 4°C) and pre-cleared with un-conjugated beads. In parallel, 2 μg of antibody was conjugated to Dynabeads for one hour at 4°C with rocking. Lysate was then added to the conjugated beads for 2 hr. Beads were then washed 5 × 500 μL of lysis buffer before being eluted in Laemmli buffer at 85°C for ten minutes.

## Bimolecular fluorescence complementation assay

Bimolecular fluorescence complementation (Split-YFP assay) was performed as previously described with some modifications (*Lee et al., 2011*). N-terminal or C-terminally tagged α-Syn and tau pBABE retroviral constructs were generated containing either the N-terminal or C-terminal portion of YFP. Viruses were generated from these constructs and stable cell lines (using HeLa cells, infected with constructs and selected with G418 for over a week) were created for both α-Syn and tau using the N-terminal portion of YFP, termed 'Bait'. Once generated, these cell lines ($5 \times 10^4$ cells) were transfected with 250 ng of 'Prey' plasmid (tau, MSK1 and TRIM28 containing the C-terminal portion of YFP). Cells were monitored for fluorescence for 48–72 hr (fluorescence begins around 24–30 hr) and cells were either collected for flow cytometry analysis (at 48 hr, using a BD LSR Fortessa with HTS [high throughput sampler] module) or fixed for fluorescence microscopy. Cells visualized for fluorescence microscopy were permeabilized and stained for DAPI to visualize nuclei and epifluorescence from the complementation was monitored.

## Quantification of nuclear translocation (IF)

The relative amount of nuclear versus cytoplasmic stained protein (α-Syn or tau) was performed in primary granule neurons previously infected (for 7 days) with lentiviral constructs overexpressing mCherry, TRIM28:mCherry or TRIM28-Mut:mCherry on coverslips. The nucleus was outlined using DAPI and the cytoplasmic outline was determined by DIC (not shown). For each biological replicate, at least 15 cells were sampled and the nuclear fraction of each cell was calculated as the intensity of the nuclear localized fluorophore divided by the intensity of the fluorophore spanning the cytoplasm and the nucleus (total intensity) and multiplied by 100. Three independent coverslips per condition

were tested and statistics were based on these three biological replicates. Alternatively, postmortem medial frontal gyrus tissue was stained for α-Syn or tau together with TRIM28 and nuclei were visualized using DAPI. Positive nuclear co-localization was set as a threshold of staining co-localizing with DAPI. At least 150 cells over the span of 5 fields (at 40x magnification) were counted per human case.

## Subcellular fractionation assay

Nuclear/Cytoplasmic fractionation was carried out in freshly spun down cells 7 days following infection (or alternatively in frozen pre-weighed post-mortem samples). Briefly, the NE-PER Nuclear and Cytoplasmic Extraction Kit (Thermo Scientific, Product #78835) was used to isolate nuclear and cytoplasmic fractions of primary neurons or tissue. Of note, the protocol of the NE-PER kit was modified to incorporate an additional PBS-wash of the nuclear pellet following the cytosolic extraction as we found it increased purity of the preparation. Probing with antibodies specific to each compartment (i.e. Lamin A and histone H3 for the nuclear fraction and Vinculin and GAPDH for the cytoplasmic fraction) was used as a measure of extraction purity and for normalization.

## Post-mortem tissue from human cases

Tissue from patients with PD, AD, PSP and control subjects were obtained from the Neuropathology Core at the Johns Hopkins Udall Centre. Tissue was obtained from consenting donors and use conformed to JHMI Institutional Review Board approved protocols. Neuropathological assessment conformed to the National Institute on Aging-Reagan consensus criteria and similar post-mortem intervals were used between samples. For this study, tissues from the frontal cortex (for PD, AD and their respective controls) or pons (for PSP and their respective controls) was utilized. Please see *Supplementary file 3* for full demographics and pathological analysis. Frozen tissue was either fractionated as per the protocol above or was alternatively processed for biochemical fractionation. In the latter case, each sample was homogenized in RIPA buffer containing protease inhibitors using a dilution of brain: RIPA of 1:10 (w/v). Samples were then centrifuged at 10,000 rpm for 20 min at 4°C. The supernatants were portioned into aliquots, snap-frozen, and stored at –80°C. The pellets were then resuspended in 88% formic acid to solubilize aggregate/insoluble protein for one hour at room temperature. Samples were then diluted to 22% in PBS and lyophilized overnight using a Savant automatic environmental speedvac system (Aes1010). Pellets were then solubilized in the same volume as the original RIPA volume and subsequently sonicated at output 2 for 30 cycles on a Branson sonicator. Independent blocks of frozen tissue were cut from PD, AD and control subjects for immunofluorescence analysis (10 µm sections).

## Statistical analyses

Experimental analysis and data collection were performed in a blinded fashion whenever possible. The sample size was chosen based on previous studies using the models described in the study in order to ensure adequate statistical power. Pre-determined exclusion criteria were used in the nuclear/cytoplasmic biochemical fractionation by determining purity of the prep. If the fractionation was not adequate (as per Lamin A, Histone H3, GAPDH and Vinculin levels), sample was re-fractionated or omitted from the study. Tissue sections with high background staining/staining artifacts were removed and/or re-stained for the analysis. Randomization together with blinding was used for histological quantification as well as human case studies. Randomization was done using Microsoft Excel and blinding was performed by assigning new non-descript codes to cases. Unblinding was only done following analysis. Outliers for histology were removed if there was a lack of detectable viral expression in the animal and if the sample met the Grubb's outlier test p-value. p values were determined using the appropriate statistical method via GraphPad Prism, as described throughout the manuscript. For simple comparisons, two-tailed Student's *t*-test was used whereas for multiple comparisons, ANOVA followed by the appropriate *post hoc* analysis were utilized. The summary of all statistical analysis is presented in *Supplementary file 1*. All data presented are of mean + (or ±) s.e.m. *, ** and *** denote $p < 0.05$, $p < 0.01$ and $p < 0.001$, respectively. ns denotes $p > 0.05$.

## Acknowledgements

The authors wish to thank the families of all patients who agreed to donate post-mortem tissue. The authors thank Drs. T Südhof (HHMI, Stanford University), M-F Chesselet (University of California Los Angeles), J Burré (Weill Cornell Medical College), J Noebels (Baylor College of Medicine), F Zhu (Florida State University), G Salvesen (Sanford Burnham Medical Research Institute), P Tsoulfas (University of Miami) and J Yuan (Harvard Medical School) for providing plasmids and animals used in this study; Dr. Olga Pletniková from the Johns Hopkins Clinical and Neuropathology Core for collecting postmortem tissue samples; and Vicky Brandt, Laura Lavery, Gabriel Vasquez-Velez, Asante Hatcher, Laura Lombardi, Sishir Subedi, Steven Baker, Vitaliy Bondar, Xiuyun Liu, and members of the Zoghbi lab for technical assistance, valuable insight and comments on the manuscript. This research was supported in part by the Michael J. Fox Foundation (target validation program 2014), the Robert A and Renée E Belfer Family Foundation, the Huffington Foundation, the Hamill Foundation, and the Howard Hughes Medical Institute (HYZ), the Canadian Institutes of Health Research Fellowship 201210MFE-290072–173743 (MWCR), the NIH/NINDS 3R01 NS027699-25S1 and 1K22NS092688-01 (CALR), the Gordon and Mary Cain Pediatric Neurology Research Foundation Laboratories, the Cell Based Assay Screening Service Core at BCM, and the confocal core at the BCM Intellectual and Developmental Disabilities Research Center (NIH U54 HD083092 from the Eunice Kennedy Shriver National Institute of Child Health and Human Development). PDD, AD, PSP and control tissues for this research were provided by the Johns Hopkins University Morris Udall Parkinson's Disease Center of Excellence (NINDS P50 NS38377) and Alzheimer's Disease Research Center (NIA P50 AG05146).

## Additional information

### Competing interests

HYZ: Senior editor, *eLife*. The other authors declare that no competing interests exist.

### Funding

| Funder | Grant reference number | Author |
|---|---|---|
| Canadian Institutes of Health Research | 201210MFE-290072-173743 | Maxime WC Rousseaux |
| National Institutes of Health | 1K22NS092688-01 | Cristian A Lasagna-Reeves |
| National Institutes of Health | 3R01 NS027699-25S1 | Cristian A Lasagna-Reeves |
| National Institutes of Health | P50 NS38377 | Juan C Troncoso |
| National Institutes of Health | P50 AG05146 | Juan C Troncoso |
| Howard Hughes Medical Institute | | Huda Y Zoghbi |
| Michael J. Fox Foundation for Parkinson's Research | Target Validation Program 2014 | Huda Y Zoghbi |
| National Institutes of Health | U54 HD083092 | Huda Y Zoghbi |

The funders had no role in study design, data collection and interpretation, or the decision to submit the work for publication.

### Author contributions

MWCR, Conceived the study, Designed experiments, Analyzed and interpreted the data, Wrote the manuscript, Performed molecular and biochemical experiments, Performed the cell-based screens, Performed mouse genotyping, surgery, histology and microscopy; MdH, Performed the Drosophila experiments, Conception and design, Analysis and interpretation of data; CAL-R, Performed mouse genotyping, surgery, histology and microscopy, Aided in data interpretation and helped in editing the manuscript; ADM, Performed the cell-based screens, Analysis and interpretation of data; JP, TJK, Provided tools and reagents; PJ-N, NL, Performed the cell-based screens; IA-R, Performed the Drosophila experiments, Analysis and interpretation of data; AS, LS, LV-V, Performed mouse genotyping, surgery, histology and microscopy; TFW, Provided reagents and aided in data interpretation;

JCT, Performed human pathological analysis and provided tissue; JB, Provided reagents and aided in data interpretation, Conception and design; HYZ, Conceived the study, Designed experiments, Analyzed and interpreted the data, Wrote the manuscript, Contributed unpublished essential data or reagents

### Author ORCIDs
Huda Y Zoghbi, http://orcid.org/0000-0002-0700-3349

### Ethics

Human subjects: Tissue from patients with PD, AD, PSP and control subjects were obtained from the Neuropathology Core at the Johns Hopkins Udall Centre. Tissue was obtained from consenting donors and use conformed to JHMI Institutional Review Board approved protocols.

Animal experimentation: Up to five mice were housed per cage and kept on a 12 h light; 12 h dark cycle and were given water and standard rodent chow ad libitum. All procedures carried out in mice were approved by the Institutional Animal Care and Use Committee for Baylor College of Medicine and Affiliates.

## Additional files

### Supplementary files

• Supplementary file 1. Summary of all statistics used throughout the manuscript. Statistical summary of each figure, including comparisons between each group. Figure in question is in column A. Assay and quantitative measurement from that figure are in columns B and C. Statistical test employed and sample size ($n$) are in D and E. Statistical comparison is in F. F values, t values (when applicable) and degrees of freedom are present in column G. Calculated p value is presented in column H.

• Supplementary file 2. Summary of reagents used throughout the manuscript. First tab shows antibodies used including concentration and application. Second tab shows shRNA sequences used. Third tab shows Plasmids used and cloning approach if applicable. Fourth tab shows qRT-PCR primers used.

• Supplementary file 3. Summary of post-mortem cases used. Demographics, pathological assessment, post-mortem delay and tissue studied are presented.

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
