## [Decision Letter]

Thank you for submitting your article "TRIM28 regulates the nuclear accumulation and toxicity of both α-synuclein and tau" for consideration by *eLife*. Your article has been reviewed by three peer reviewers, and the evaluation has been overseen by a Reviewing Editor. The reviewers have opted to remain anonymous.

The reviewers have discussed the reviews with one another and the Reviewing Editor has drafted this decision to help you prepare a revised submission.

Summary:

In a shRNA screen, Rousseaux et al. identify TRIM28 as a regulator of α-synuclein and tau, two proteins involved in neurodegeneration. The authors' data suggests that deceasing levels of TRIM28 decreases levels of these two proteins and their toxicity, and conversely increasing levels has the opposite consequences. The authors go on to further investigate the role of TRIM28 on the intracellular localization of synuclein and tau, potential physical interactions, and potential effects of TRIM28 on the half-lives of these proteins. Overall the manuscript presents interesting and novel data.

Essential revisions:

1) In several places the addition of experimental details would greatly enhance the readability of the manuscript. Several places where details would help are listed in individual reviews.

2) The authors show data that TRIM28 acts on synuclein and tau stabilization. However, there is much literature on other functions of TRIM28 in gene regulation and epigenetic control. Data from experiments described in this manuscript could be consistent with an indirect function of TRIM28 on synuclein and tau. This should be acknowledged and commented on.

3) The half-life experiment examining TRIM28's effect on synuclein and tau should examine transcript levels of these genes in this experiment (and in experiments utilizing primary neurons or transgenic mouse brains). Since vinculin is used as a control, it's half-life should also be described. Also how do the author's rule out that translational changes (that do not have to do with protein half-life) are not affected by TRIM28.

4) In terms of determining whether the ligase activity of TRIM28 is necessary for nuclear localization of tau and synuclein the authors examine mutant TRIM28. These results need to be interpreted in light of whether the mutant protein still interacts with tau and synuclein.

5) Finally, the authors suggest that changes in TRIM28 levels and localization may be a "driving force" in nuclear accumulation of synuclein and tau. However, the reviewers suggest that this data does not suggest causality. Additional important questions regarding this experiments (see Reviewer 2 comments) should be addressed and the conclusions from TRIM28 localization changes in diseased human brain need to be reinterpreted in light of reviewer's comments.

Reviewer #1:

In a shRNA screen Rousseaux et al. found that TRIM28 regulate the localization of α-synuclein and tau. Further, the authors demonstrated that TRIM28 regulates the toxicity of these proteins in both the fly eye and mouse brain. Although the manuscript fails to elucidate mechanism, given that α-synuclein and tau are associated with neurodegenerative disorders, these findings are of interest to the neurodegeneration field. However additional data would strengthen the authors' findings. For example, the authors have obtained a conditional allele of Trim 28, what is the effect of complete deletion on the pathology associated with α-synuclein or tau expression. Does deletion of Trim28 have additional actions on the transcriptional profile that could secondarily affect stability of α-synuclein and tau?

Using a Dox inducible cell line of α-synuclein and tau, the authors suggest that Trim 28 changes the stability of these proteins. Do the transcript levels change with Trim28 changes? What is the effect of Trim28 on endogenous levels of α-synuclein and tau. Is there a change in localization of these proteins in these cells which could explain the difference in protein decay?

Other points:

Overall the manuscript would benefit with more description of experiments within the text (this reviewer had to look in legends and in methods often). Examples are given below.

Figure 1. What primary neurons were used? Do the dots in Figure 1 (and other bar graphs) represent individual experiments?

Figure 2. In the right 2 panels is a tau/GFP transgene expressed (versus just tau)? The reference for the tau-transgenic *Drosophila* should be included in the text of the results. In panel B, were an n=15 used for each experiment and were there two experiments performed? What tissue is used in panel C? How much of a "partial" loss of function are the two Trim28 alleles i.e., how much wild type Trim28 remains?

Figure 3. A sentence explaining the α-Syn overexpressing mice and an explanation on why serine 129-phosphorylated α-Syn and hippocampus (instead of SN) was would be helpful for the reader.

Figure 4. CA1 thickness was measured and shown to decrease in when Trim28 was overexpressed in α-Syn and tau transgenic mice. Does the neuron number change? Are dying neurons observed?

Figure 5. An explanation of the pINDUCER20 cell lines would be helpful for the reader. What cells were these experiments performed in?

Figure 6. The authors perform IP and BiFC to ask whether Trim28 directly binds α-Syn and tau. However, these approaches do not test direct interactions. Do bacterial recombinant proteins interact? Does the C65A/C68A mutant Trim28 still form a complex with α-Syn and tau?

Reviewer #2:

The current study by Rousseaux et al. identifies a novel regulator of both α-Synuclein and Tau protein levels, TRIM28. Using a number of in vitro and in vivo models the authors convincingly show that knock-down of TRIM28 is sufficient to reduce α-Synuclein and Tau levels and their associated neurotoxicity. Conversely, overexpression of TRIM28 increases α-Synuclein and Tau half-lives, and accordingly further exacerbates their neurotoxicity in vivo. Overall the authors are commended for very nicely executed, elegant experiments particularly in vitro and in the animal models and this work is of sufficient novelty and interest for a broad audience in the neurodegenerative disease field.

That being said, we have reservations regarding the interpretation of the patient data presented in Figure 7, Figure 7—figure supplement 1 and Figure 7—figure supplement 2, where the authors conclude that aberrant TRIM28 underlie disease pathologies. Overall, looking at the biochemical data we do not see a clear increase in overall TRIM28 levels under disease conditions. The authors indeed find a statistically significant shift of TRIM28 to the insoluble form in these conditions, but we wonder how TRIM28 could participate in actively bringing α-Synuclein/Tau into the nucleus if it's bound up in aggregates, presumable inactive. By staining (Figure 7), the nuclear localization of TRIM28 in both control and disease patient brain also makes us wonder how TRIM28 would access these cytosolic proteins to bring them into the nucleus. Moreover, when looking at individual human samples in the biochemistry provided in Figure 7—figure supplement 1, there is no clear correlation between TRIM28 levels and α-Synuclein/Tau levels; For example, in Figure 7—figure supplement 1, Panel C, right, the up/down levels of TRIM28 in the insoluble fraction does not correlate with the increase/decrease of phospho-Tau levels of the same patient. Perhaps in an earlier disease stage the data would make more sense. Overall it seems the human data is not as black-and-white as the more conclusive animal model data presented in Figure 1–Figure 6 and the manuscript should be modified to make the conclusions more modest.

Reviewer #3:

In this manuscript the authors report the results of a focused genetic screen looking for genes that co-regulate levels of tau and synuclein, proteins that contribute to multiple neurodegenerative diseases, and the identification of TRIM28. Decreasing levels of TRIM28 decreases tau and synuclein levels and toxicity, while overexpression has the opposite effect. The focus is on possible direct physical interactions that might mediate localization of tau and synuclein to the nucleus. Overall the findings are interesting and novel.

1) The authors tested changes in levels of other degeneration-causing proteins and did not see any change. They conclude that TRIM28 is likely to be a key postranslational regulator of tau and synuclein. However, the list of proteins tested is small. There is a rather large literature on TRIM28 and its roles in multiple cellular processes. TRIM28 has also been shown to be a global regulator of many aspects of gene regulation and epigenitic control. I would encourage the authors to consider and comment on these activities and the possibility that effects on tau and synuclein might have an indirect component.

2) The authors may want to define the term "druggable", as the definition may be different depending on the investigator.

3) For the fly experiments involving tau overexpression in the eye, it would be useful to know that there are no effects on other GMR promoter-dependent phenotypes. This would act as a stand in for changes in transcriptional regulation. The climbing assay has similar issues in that a specific promoter is being used and TRIM28 is known to have effects on transcriptional regulation.

4) Next, they moved to mice overexpressing synuclein in dopaminergic neurons. This experiment has a nice control in that there is a GFP included as an IRES element. This figure (3) shows that while GFP levels remain constant, synuclein levels decrease. Importantly, the number of TH positive neurons decreases in the presence of synuclein, and this is reversed following RNAi of TRIM28. These observations also are consistent with a direct mode of action, but probably do not exclude indirect modes of action through intermediate proteins. Again, I think the authors should acknowledge this possibility.

5) Finally, the authors used a TRIM28 heterozygote to look for dose-dependent effects on phosphorylated synuclein. Can they comment on why this assay was used as opposed to that in the above panels? Is this simply more sensitive? Does it reflect a particular pathology? If so, can they provide a reference for this biology?

6) The authors did a half-life experiment suggesting that TRIM28 expression stabilizes synuclein and tau. It would be important to show that other proteins are not stabilized, and to know what Vinculin's half-life is in order for it to be a useful control. For example, if its half-life is very long, its levels would not change much even if it was being affected. Can the authors rule out effects on translation from existing transcripts? I just point this out because if translation is not inhibited (there is no evidence that cyclohexamide was used to block translation in the half life experiments), there could be effects on protein abundance over time that are translational, and would otherwise be scored as alterations in protein half-life.

7) The authors asked if TRIM28's ligase activity is required for nuclear localization of tau and synuclein. They did this by expressing a mutant version of TRIM28 that has two cysteine mutations in the ring domain. As these cysteines play a structural role in the ring domain, these mutations could disrupt the overall folding/conformation of this domain. It is important to determine if mutant versions of TRIM28 still interacts with tau and synuclein. This would be an important experiment to distinguish between ligase-dependent mechanisms of regulation versus physical binding and recruitment to the nuclear compartment. If binding is lost it would be difficult to conclude more than that interactions are important.

8) The authors saw increased levels of TRIM28 in an insoluble fraction in human disease brains and that tau and synuclein were enriched in nuclei. From this they suggested that "The aberrant change in TRIM28 levels and distribution could therefore be a driving force in the nuclear accumulation of α-Syn and tau. " To test this they look for colocalization of TRIM28 in nuclei of disease brains and see significantly more cells that have colocalization of synuclein, tau and TRIM28 than in controls. This result is interesting. But could it just be that TRIM28 becomes trapped in insoluble nuclear complexes with synuclein and tau in disease brains? I am not sure I see a link that is necessarily causal. Can the authors discuss why an increase in number of nuclei with TRIM28, synuclein and tau should be taken to indicate that TRIM28 is driving the process rather than being carried along for the ride as an interacting protein?

---

## [Author Response]

*Essential revisions:*

*1) In several places the addition of experimental details would greatly enhance the readability of the manuscript. Several places where details would help are listed in individual reviews.*

We have expanded on our description of experimental details throughout the manuscript, particularly in the Results section.

2) The authors show data that TRIM28 acts on synuclein and tau stabilization. However, there is much literature on other functions of TRIM28 in gene regulation and epigenetic control. Data from experiments described in this manuscript could be consistent with an indirect function of TRIM28 on synuclein and tau. This should be acknowledged and commented on.

We now acknowledge and comment on a putative indirect mechanism in the Discussion section.

*3) The half-life experiment examining TRIM28's effect on synuclein and tau should examine transcript levels of these genes in this experiment (and in experiments utilizing primary neurons or transgenic mouse brains). Since vinculin is used as a control, it's half-life should also be described. Also how do the author's rule out that translational changes (that do not have to do with protein half-life) are not affected by TRIM28.*

We repeated the stability assay and monitored levels of *SNCA* and *MAPT* transcripts (Figure 5—figure supplement 1). We find that TRIM28 overexpression does not alter RNA decay kinetics thus suggesting that TRIM28 acts post-translationally on protein stability. While we did not test the effect of TRIM28 overexpression on *Snca* and *Mapt* in primary neurons, we have looked at their transcripts in mice lacking a copy of Trim28 and do not see any changes in their transcript levels (Figure 1—figure supplement 3, lower panel) suggesting again that TRIM28 acts on the protein levels of α-Syn and tau, not their DNA or RNA. Moreover, we wish to clarify that, since this is a doxycycline-inducible system (and not a cycloheximide assay) we only affect the expression of α-Syn and tau under the control of a doxycycline-inducible element, therefore vinculin levels (and total protein levels for that matter) do not change during the time course.

*4) In terms of determining whether the ligase activity of TRIM28 is necessary for nuclear localization of tau and synuclein the authors examine mutant TRIM28. These results need to be interpreted in light of whether the mutant protein still interacts with tau and synuclein.*

We tested the binding affinity of α-Syn and tau for TRIM28 in the context of this E3-ligase mutant and found that, while TRIM28 E3-ligase mutant was itself less stable (as previously noted in Figure 6—figure supplement 1), its binding to α-Syn and tau was not abolished. This suggests that the loss of nuclear localization seen when comparing TRIM28-WT to TRIM28-Mut is at least partly due to a loss of function in addition to this structure disruption. These results are now presented as Figure 6—figure supplement 2.

*5) Finally, the authors suggest that changes in TRIM28 levels and localization may be a "driving force" in nuclear accumulation of synuclein and tau. However, the reviewers suggest that this data does not suggest causality. Additional important questions regarding this experiments (see Reviewer 2 comments) should be addressed and the conclusions from TRIM28 localization changes in diseased human brain need to be reinterpreted in light of reviewer's comments.*

We agree that human data from post mortem tissue rarely suggests causality (as it is just a snapshot of time, little can be definitively concluded) and have thus toned down the interpretation in the Results and Discussion.

*Reviewer #1:*

*In a shRNA screen Rousseaux et al. found that TRIM28 regulate the localization of α-synuclein and tau. Further, the authors demonstrated that TRIM28 regulates the toxicity of these proteins in both the fly eye and mouse brain. Although the manuscript fails to elucidate mechanism, given that α-synuclein and tau are associated with neurodegenerative disorders, these findings are of interest to the neurodegeneration field. However additional data would strengthen the authors' findings. For example, the authors have obtained a conditional allele of Trim 28, what is the effect of complete deletion on the pathology associated with α-synuclein or tau expression. Does deletion of Trim28 have additional actions on the transcriptional profile that could secondarily affect stability of α-synuclein and tau?*

We thank the reviewer for their constructive feedback. Given that complete deletion of TRIM28 is embryonic lethal, we employed Trim28 heterozygous mice as a model to genetically control for TRIM28 levels. Although conditional null mice have transcriptional changes (Jakobsson et al., 2008), we suspect that a heterozygous mouse should have much less effects on global transcriptional changes, though we cannot rule this out without performing a large scale profiling experiment on the brain which is beyond the scope of this study.

*Using a Dox inducible cell line of α-synuclein and tau, the authors suggest that Trim 28 changes the stability of these proteins. Do the transcript levels change with Trim28 changes? What is the effect of Trim28 on endogenous levels of α-synuclein and tau. Is there a change in localization of these proteins in these cells which could explain the difference in protein decay?*

As per this reviewer’s suggestion, we examined *SNCA* and *MAPT* transcript levels in the Dox-inducible cell line in response to TRIM28 expression. We found that the RNA decay rates were not altered upon TRIM28 overexpression suggesting that TRIM28 acts post-translationally (Figure 5—figure supplement 1). Moreover, we show that TRIM28 increases the endogenous levels of α-Syn and tau in primary cerebellar granule neurons (Figure 1—figure supplement 3). We did not test whether the change in localization of these proteins affect protein decay rates but propose that the difference in protein decay may be attributed to nuclear-localized α-Syn or tau escaping their native quality control mechanism(s).

*Other points:*

*Overall the manuscript would benefit with more description of experiments within the text (this reviewer had to look in legends and in methods often). Examples are given below.*

We have clarified certain experiments throughout the text. We have answered these point-by-point below.

*Figure 1. What primary neurons were used? Do the dots in Figure 1 (and other bar graphs) represent individual experiments?*

We used primary cerebellar granule neurons from P7-9 pups. This has been clarified throughout the text and in the Methods section. Each dot in the bar graphs represents an individual experiment (i.e. individual culture or animal).

*Figure 2. In the right 2 panels is a tau/GFP transgene expressed (versus just tau)? The reference for the tau-transgenic Drosophila should be included in the text of the results. In panel B, were an n=15 used for each experiment and were there two experiments performed? What tissue is used in panel C? How much of a "partial" loss of function are the two Trim28 alleles i.e., how much wild type Trim28 remains?*

The genotypes for the panels in Figure 2 are specified in [Supplementary-material SD3-data], tab “Fly lines”. The allele for *Drosophila* dsTrim28 is under the control of a UAS promoter and is expressed using the Gal4/UAS system. To avoid potential artifacts due to titration of Gal4 we routinely use a neutral UAS (UAS-eGFP), in order to achieve similar expression of tau in both the control (UAS-tau/UAS-GFP) and the experimental (UAS-tau/UAS-dsTrim28).

These tau *Drosophila* transgenic lines have not been previously published. A detailed description of how we generated them is included in the *Drosophila* Methods under the “Materials and methods” section as a “human wild-type 4 repeat tau under the control of a UAS promoter”.

The legend for Figure 2 has been modified to include number of experiments and tissue used in panel C (adult *Drosophila* retina expressing tau/GFP or tau/dsTrim28).

The description of the Trim28 alleles used is in the Methods section. Trim28^LOF-1^ is a loss of function allele of bonus because it fails to complement Trim28^LOF-2^, which has already been described as a bonus null allele (Beckstead et al., 2001). These alleles show a 50% reduction in heterozygosis.

*Figure 3. A sentence explaining the α-Syn overexpressing mice and an explanation on why serine 129-phosphorylated α-Syn and hippocampus (instead of SN) was would be helpful for the reader.*

We have clarified this in the Results section. Both native and transgenic α-Syn are highly expressed in the hippocampus and its clear phosphorylation in this location is more readily quantifiable (Chesselet et al., 2012).

*Figure 4. CA1 thickness was measured and shown to decrease in when Trim28 was overexpressed in α-Syn and tau transgenic mice. Does the neuron number change? Are dying neurons observed?*

We cannot definitively say that there is neuronal loss following Trim28 overexpression as NeuN staining on paraffin sections hasn’t worked nicely in our hands. That said, previous reports suggest that NeuN loss in this mouse model is accompanied by astrocyte accumulation (Dhungel et al., 2015). Given that we observe an increase in astrocytic number in the CA1 (with a concomitant decrease in CA1 thickness), this would be suggestive of cell loss. We have not tested whether neurons themselves are dying as the mechanism through which α-Syn and tau cause the demise of neurons (i.e. apoptosis, autophagy, etc.) in these models is not entirely clear.

*Figure 5. An explanation of the pINDUCER20 cell lines would be helpful for the reader. What cells were these experiments performed in?*

We have clarified this in the Results section to explain the system in more detail. SH-SY5Y cells stably expressing the pINDUCER constructs (G418 selected) were used.

*Figure 6. The authors perform IP and BiFC to ask whether Trim28 directly binds α-Syn and tau. However, these approaches do not test direct interactions. Do bacterial recombinant proteins interact? Does the C65A/C68A mutant Trim28 still form a complex with α-Syn and tau?*

We agree that these results suggest that TRIM28 forms a complex with α-Syn and tau but does not mean that they directly interact. While we have had difficulty in purifying recombinant TRIM28 to test the direct interaction, we have now tested the E3-ligase mutant in the context of the immunoprecipitation experiment. We find that, while the C65A/C68A mutant is less stable, it can still form a complex with α-Syn and tau (Figure 6—figure supplement 2).

*Reviewer #2:*

*et al.α*in vitroin vivo*αα*in vivoin vitro*[…]Overall, looking at the biochemical data we do not see a clear increase in overall TRIM28 levels under disease conditions. The authors indeed find a statistically significant shift of TRIM28 to the insoluble form in these conditions, but we wonder how TRIM28 could participate in actively bringing α-Synuclein/Tau into the nucleus if it's bound up in aggregates, presumable inactive. By staining (Figure 7), the nuclear localization of TRIM28 in both control and disease patient brain also makes us wonder how TRIM28 would access these cytosolic proteins to bring them into the nucleus. Moreover, when looking at individual human samples in the biochemistry provided in Figure 7—figure supplement 1, there is no clear correlation between TRIM28 levels and α-Synuclein/Tau levels; For example, in Figure 7—figure supplement 1, Panel C, right, the up/down levels of TRIM28 in the insoluble fraction does not correlate with the increase/decrease of phospho-Tau levels of the same patient. Perhaps in an earlier disease stage the data would make more sense. Overall it seems the human data is not as black-and-white as the more conclusive animal model data presented in Figure 1–Figure 6 and the manuscript should be modified to make the conclusions more modest.*

We thank the reviewer for their appreciation of this work and its impact. We agree that pathological analysis of human tissue data in the context of TRIM28 can be difficult to interpret and have thus toned down our conclusions to make them more modest (Results and Discussion).

*Reviewer #3:*

[…]

*1) The authors tested changes in levels of other degeneration-causing proteins and did not see any change. They conclude that TRIM28 is likely to be a key postranslational regulator of tau and synuclein. However, the list of proteins tested is small. There is a rather large literature on TRIM28 and its roles in multiple cellular processes. TRIM28 has also been shown to be a global regulator of many aspects of gene regulation and epigenitic control. I would encourage the authors to consider and comment on these activities and the possibility that effects on tau and synuclein might have an indirect component.*

We now comment more on the potential indirect effect of TRIM28 on α-Syn and tau in the Discussion section.

2) The authors may want to define the term "druggable", as the definition may be different depending on the investigator.

We have clarified our definition for “druggable” in the first paragraph of the Results section.

*3) For the fly experiments involving tau overexpression in the eye, it would be useful to know that there are no effects on other GMR promoter-dependent phenotypes. This would act as a stand in for changes in transcriptional regulation. The climbing assay has similar issues in that a specific promoter is being used and TRIM28 is known to have effects on transcriptional regulation.*

To address the reviewers concern we have performed qPCR on animals carrying Trim28 LOF alleles used in this study. All the results show that partial knockdown of Trim28 does not decrease the levels of Gal4 (Figure 2—figure supplement 2).

*4) Next, they moved to mice overexpressing synuclein in dopaminergic neurons. This experiment has a nice control in that there is a GFP included as an IRES element. This figure (3) shows that while GFP levels remain constant, synuclein levels decrease. Importantly, the number of TH positive neurons decreases in the presence of synuclein, and this is reversed following RNAi of TRIM28. These observations also are consistent with a direct mode of action, but probably do not exclude indirect modes of action through intermediate proteins. Again, I think the authors should acknowledge this possibility.*

Please see answer to point #1.

*5) Finally, the authors used a TRIM28 heterozygote to look for dose-dependent effects on phosphorylated synuclein. Can they comment on why this assay was used as opposed to that in the above panels? Is this simply more sensitive? Does it reflect a particular pathology? If so, can they provide a reference for this biology?*

We used this approach as a complementary approach to the dual-virus approach. This animal model recapitulates certain facets of PD that are not seen in the dual virus model such as pathological phosphorylation of α-syn. As such, we wanted to test whether reduction of TRIM28 could not only decrease cell toxicity but also pathological phosphorylation of α-syn. We have now referenced this model in more detail in the Results section as well as in the Materials and methods.

*6) The authors did a half-life experiment suggesting that TRIM28 expression stabilizes synuclein and tau. It would be important to show that other proteins are not stabilized, and to know what Vinculin's half-life is in order for it to be a useful control. For example, if its half-life is very long, its levels would not change much even if it was being affected. Can the authors rule out effects on translation from existing transcripts? I just point this out because if translation is not inhibited (there is no evidence that cyclohexamide was used to block translation in the half life experiments), there could be effects on protein abundance over time that are translational, and would otherwise be scored as alterations in protein half-life.*

We have now tested RNA decay kinetics in the pINDUCER system following TRIM28 overexpression. We find that *SNCA* and *MAPT* RNA decay is unaffected by TRIM28 overexpression status. These data are now presented as Figure 5—figure supplement 1. In this inducible system, like all the other systems tested, vinculin levels did not change following dox addition or TRIM28 overexpression.

*7) The authors asked if TRIM28's ligase activity is required for nuclear localization of tau and synuclein. They did this by expressing a mutant version of TRIM28 that has two cysteine mutations in the ring domain. As these cysteines play a structural role in the ring domain, these mutations could disrupt the overall folding/conformation of this domain. It is important to determine if mutant versions of TRIM28 still interacts with tau and synuclein. This would be an important experiment to distinguish between ligase-dependent mechanisms of regulation versus physical binding and recruitment to the nuclear compartment. If binding is lost it would be difficult to conclude more than that interactions are important.*

We thank the reviewer for the good suggestion. We have tested the interaction of α-Syn and tau with either wild type TRIM28 or E3 ligase mutant TRIM28. We find that, while the TRIM28 E3 mutant itself is less stable, it can still interact with α-Syn and tau thus suggesting that the nuclear localization is at least partly mediated through its E3 ligase activity (Figure 6—figure supplement 2).

*8) The authors saw increased levels of TRIM28 in an insoluble fraction in human disease brains and that tau and synuclein were enriched in nuclei. From this they suggested that "The aberrant change in TRIM28 levels and distribution could therefore be a driving force in the nuclear accumulation of α-Syn and tau. " To test this they look for colocalization of TRIM28 in nuclei of disease brains and see significantly more cells that have colocalization of synuclein, tau and TRIM28 than in controls. This result is interesting. But could it just be that TRIM28 becomes trapped in insoluble nuclear complexes with synuclein and tau in disease brains? I am not sure I see a link that is necessarily causal. Can the authors discuss why an increase in number of nuclei with TRIM28, synuclein and tau should be taken to indicate that TRIM28 is driving the process rather than being carried along for the ride as an interacting protein?*

We agree that post-mortem data only offers a snapshot of time and cannot suggest causality. Thus, per the suggestions of reviewers #1 and 2, we have toned down our claim that TRIM28 drives the nuclear localization of α-Syn and tau in human disease.

References:

Jakobsson, J., Cordero, M.I., Bisaz, R., Groner, A.C., Busskamp, V., Bensadoun, J.C., Cammas, F., Losson, R., Mansuy, I.M., Sandi, C.*, et al.* (2008). KAP1-mediated epigenetic repression in the forebrain modulates behavioral vulnerability to stress. Neuron *60*, 818-831.